# Synergistic cooperation promotes multicellular performance and unicellular free-rider persistence

William W. Driscoll[1,2] & Michael Travisano[1,2]

The evolution of multicellular life requires cooperation among cells, which can be undermined by intra-group selection for selfishness. Theory predicts that selection to avoid non-cooperators limits social interactions among non-relatives, yet previous evolution experiments suggest that intra-group conflict is an outcome, rather than a driver, of incipient multicellular life cycles. Here we report the evolution of multicellularity via two distinct mechanisms of group formation in the unicellular budding yeast *Kluyveromyces lactis*. Cells remain permanently attached following mitosis, giving rise to clonal clusters (staying together); clusters then reversibly assemble into social groups (coming together). Coming together amplifies the benefits of multicellularity and allows social clusters to collectively outperform solitary clusters. However, cooperation among non-relatives also permits fast-growing unicellular lineages to 'free-ride' during selection for increased size. Cooperation and competition for the benefits of multicellularity promote the stable coexistence of unicellular and multicellular genotypes, underscoring the importance of social and ecological context during the transition to multicellularity.

[1] The Biotechnology Institute, University of Minnesota, 1479 Gortner Avenue, St Paul, Minnesota 55108, USA. [2] Department of Ecology, Evolution and Behavior, University of Minnesota, 100 Ecology Building, 1987 Upper Buford Circle, Roseville, Minnesota 55108, USA. Correspondence and requests for materials should be addressed to W.W.D. (email: wwdriscoll@gmail.com).

Cooperation is a defining feature of living systems and is integral to biological function across levels of organization, including genomes, multicellular organisms and societies[1]. However, the prevalence and success of cooperation is surprising considering the tendency for direct competition to favour selfishness, even to the point of compromising biological functionality[2–4]. Substantial effort has been dedicated to understanding the processes underlying both the successes and failures of cooperation. The most influential framework, kin selection theory, focuses on genetic relatedness as a means of promoting the evolution and maintenance of cooperation within groups of close relatives[5]. According to kin selection, the cost of cooperation can be offset by benefits to relatives, which are more likely to share alleles associated with cooperation[6–8].

The origin and maintenance of cooperation among formerly free-living unicells is often viewed as a barrier for the evolution of complex multicellular life[1,9,10]. This major evolutionary transition has occurred several times in disparate lineages, showing that solutions to evolving cooperation have evolved multiple times[11]. Evolutionary models suggest that cooperation depends on high relatedness among cells within a multicellular group; conversely, genetic diversity promotes competition, which ultimately favours selfish lineages that benefit from cooperation without reciprocating (cheaters[12]). Selection among multicellular collectives should therefore favour those lineages best able to sustain high levels of cooperation by minimizing the accumulation of internal genetic diversity[13–15]. Nevertheless, cooperation among non-relatives has been documented multiple times[16,17], including altruism during the development of multicellular chimeras[18]. The extent to which the success of incipient multicellular cooperation depends on mechanisms of avoiding nonkin therefore remains contentious[19].

The means by which multicellular groups are formed plays a central role in determining the frequency and strength of interactions among non-relatives. Initially solitary cells may form multicellular groups through two fundamentally distinct mechanisms: continued association of daughter cells following division or aggregation of pre-existing cells[12] (staying together (ST) and coming together (CT), respectively[20]). Many, if not most, microbes are capable of simple forms of multicellularity using one or both mechanisms[21]. However, all of the largest and most complex multicellular organisms rely on ST following cellular mitosis[22]. This striking disparity is widely viewed as a reflection of the divergent opportunities for intra-group genetic conflict: ST results in clonal groups, whereas groups formed by CT may harbour multiple lineages, including cheaters.

Experimental evolution provides a means of directly observing the first steps in the transition to multicellularity from unicellular ancestors. Previous evolution experiments have supported the expectation that incipient multicellular life cycles profoundly influence the stability of cooperation. Cheaters that arise through mutation proliferate and ultimately undermine multicellular cooperation in large, long-lived bacterial mats formed by ST[23], whereas frequent genetic bottlenecks can preclude cheating in smaller and shorter-lived eukaryotic groups formed by ST (in both yeast[24,25] and algae[26]). Evolution experiments with a cellular slime mould found that cheaters proliferated and disrupted cooperative development of groups formed by CT, whereas experimentally imposed unicellular bottlenecks (intended to simulate ST) were sufficient to stabilize cooperation[14]. However, systematic comparisons among these disparate outcomes remain difficult due to major differences in the features of unicellular ancestors and experimental conditions, including selection for multicellularity.

Here we report on the experimental evolution of multi-cellularity in initially unicellular populations of the budding yeast *Kluyveromyces lactis*. We identify instances of parallelism and divergence with experimentally evolved populations derived from another budding yeast, *Sachharomyces cerevisiae*[24]. Although both yeast quickly evolve parallel mechanisms of multicellular cluster formation, we observe several unique outcomes in *K. lactis* populations. We present evidence that multicellularity in *K. lactis* involves an interaction between two distinct mechanisms of group formation (corresponding to ST and CT), providing an explanation for instances of divergence with *S. cerevisiae*.

## Results

**Parallel evolution of multicellularity in divergent budding yeast.** We observed evolution in populations derived from the unicellular dairy yeast *K. lactis* maintained under conditions previously shown to favour the evolution of multicellularity in *S. cerevisiae*[24]. These two species of budding yeast are separated by ∼100 million years of evolution, including the whole genome duplication in the progenitor of *Saccharomyces*[27]. Many yeast are capable of CT through social flocculation and, although the unicellular ancestor used in the present study is considered non-flocculent, we noted very low levels of flocculation in this strain (see below).

Following the procedure of Ratcliff *et al.*[24], we subjected ten initially unicellular populations derived from *K. lactis* strain NRL Y-1140 to selection for multicellularity every 24 h (∼6.7 generations) for 60 days (∼ 400 generations). We selected for rapid sedimentation by transferring only the bottommost fraction (6.7%) of a static subculture to fresh medium following 7 min of gravitational settling ('settling selection'). The preferential survival of larger particles in this routine serves as a proxy for conditions that select for clustering or aggregation of unicells in natural communities, including size-dependent predation,[28] resource exploitation[29] and dispersal[30].

Multicellular clusters evolved in all ten *K. lactis* populations by the tenth round of settling selection (∼70 generations), but did not exclude unicellular genotypes, which persisted throughout the experiment in all replicates (Fig. 1a–c; see Supplementary Data 1 for quantitative descriptions of snowflake clusters). Multicellular isolates from the 60th transfer settled rapidly in comparison to both ancestral and co-occurring derived unicells (Fig. 1d). Multicellular *K. lactis* clusters formed through continued association of daughter cells following division: cells within clusters were attached at bud scars (Fig. 1f,g) and cluster expansion through cellular growth was apparent from time-lapse photography. Larger clusters fractured into distinct daughter clusters during growth (Supplementary Movie 1). This mode of multicellularity parallels 'snowflake' clusters previously evolved in *S. cerevisiae*[24,25]. Also as before, multicellularity is associated with reduced growth rates in *K. lactis*: multicellular isolates required more time to reach maximal growth rates (analysis of variance (ANOVA): $F(1, 18) = 32.7$, $P < 0.0001$), which were reduced compared with unicellular isolates ($F(1, 18) = 123.9$; $P < 0.0001$; Fig. 1e and Supplementary Data 2). The parallel phenotypic evolution of snowflake cluster formation indicates that this phenomenon is not contingent on the particular idiosyncracies of *S. cerevisiae*.

**Unicellular and multicellular coexistence.** The persistence of unicells in all populations for the duration of the experiment in *K. lactis* marks a notable point of divergence from previous work with *S. cerevisiae*, in which snowflakes inexorably exclude uni-cellular competitors[24]. We sought to determine the contribution of selection to the persistence of fast-growing unicellular lineages in competition with multicellular competitors. First, we tested the ability of each phenotype (unicell, snowflake) to increase from

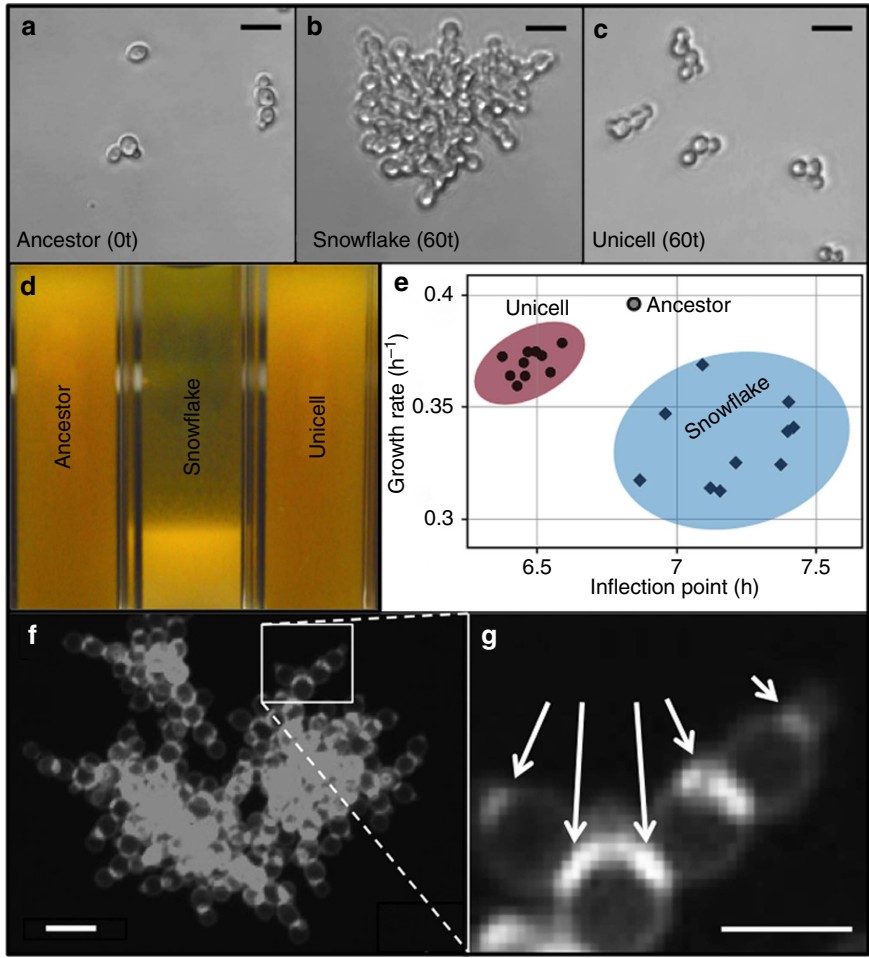

**Figure 1 | Morphological diversity of evolved *K. lactis* populations.** (**a**) Ancestral cells (strain Y-1140) typically occur as dyads or single cells, with occasional clusters of <8 cells. (**b**) All populations quickly evolved multicellular ('snowflake') strains, which occur almost exclusively as large clusters (Supplementary Data 1). (**c**) Derived unicells resemble the ancestral form and were present in all ten populations at the end of the experiment. (**d**) Visual comparison of settling in 3 ml overnight cultures after 10 m in ancestral, unicellular and snowflake isolates. (**e**) Growth parameters of ancestral (open circle; $n = 1$), derived snowflake ($n = 10$) and derived unicellular ($n = 10$) lineages. Points show mean parameter values estimated by fitting four parameter Gompertz models to each of four replicate populations per isolate. Both inflection point ($t(18.0) = -11.13$, $P < 0.0001$ and maximum growth rate are significantly different between derived snowflakes and unicells ($t(18.0) = 5.72$, $P < 0.0001$; Supplementary Data 2); shaded ellipses show 95% CI. (**f**) Calcfluor white fluorescence of a group of three snowflakes. (**g**) Cell attachment within snowflakes occurs at bud scars (areas of heightened fluorescence, indicated by white arrows), reflecting continued association of daughter cells following division. All scale bars, 10 µm.

low initial frequency (∼10% of colony-forming units (CFUs)) within a population dominated by the opposite phenotype (snowflake, unicell). We conducted two independent mutual invasion experiments using pairs of unicell and snowflake genotypes isolated from the final time point of each of the ten evolved populations. Each phenotype in every pair served as both resident and invader, totalling 20 experimental populations for each replicate experiment. Populations were subjected to settling selection for 12 days (∼80 generations). We quantified the overall performance of a focal phenotype $i$ as the log ratio of initial to final abundance (the Malthusian parameter, $m_i$). The relative fitness of $i$ is then defined as the ratio of its performance to that of its competitor, $j$ ($w_i = m_i/m_j$)[31]. To account for fluctuating selection, we quantified fitness as the geometric mean of two estimates of $w_i$, made before and after settling selection on the final day of the experiment.

Both phenotypes persisted throughout the experiments, regardless of initial frequency (Fig. 2a). A 2 × 2 ANOVA revealed main effects of initial frequency (F(1, 73) = 144.9; $P < 0.0001$) and phenotype (F(1, 73) = 13.5; $P = 0.005$). *Post-hoc* analyses using Tukey's honest significant difference (HSD) showed that selection

consistently favoured invaders over residents ($P < 0.0001$) and both phenotypes had higher fitness as invaders than as residents, consistent with stable coexistence (Tukey's HSD, $P < 0.0001$; Fig. 2b and Supplementary Data 3). Interestingly, although there was no significant fitness difference between phenotypes as residents ($P = 0.482$), snowflakes had significantly higher fitness as invaders than unicells ($P = 0.0019$).

We employed principles of community ecology to identify the mechanisms underlying unicell persistence in the face of strong selection for multicellularity. Specifically, we tracked unicell and snowflake numerical responses to settling selection, as well as the intervening 24 h of growth, to determine whether coexistence depended on fluctuating environments[32,33]. Based on differences in growth parameters (Fig. 1), we anticipated that unicellular strains would have a competitive advantage during growth and multicellular strains would have superior settling ability. Theory predicts stable coexistence when competitors have greater similarity in average relative fitness (quantified by $w$) than in ecological niche[32]. The degree of niche overlap ($\rho$) is determined by the strength of competition within and between populations (that is, the competition coefficients, $\alpha_{ij}$, where $i$ is the focal

phenotype and $j$ the competitor). We estimated competition coefficients for both phenotypes during both phases of selection by ordinary least squares regression (see Methods).

Density dependence during settling selection alone satisfies the criterion for coexistence ($\rho < w < 1/\rho$), which is notable because settling is the selective factor favouring the evolution of multicellularity in the first place (Table 1). Interestingly, unicells significantly increase during settling ($m_U = 0.39$, 95% confidence interval (95% CI) = (0.32, 0.46); ~48% increase), although this figure is modest compared with snowflakes ($m_{SF} = 1.39$, (1.33, 1.46); ~300% increase). Competition during settling selection primarily operates within phenotypes: least-squares regression revealed significant negative effects of intra-phenotype competition ($\alpha_{U,U} = -0.42$, 95% CI = (-0.71, -0.14), $P = 0.0038$; $\alpha_{SF,SF} = -1.06$, 95% CI = (-1.24, -0.89), $P < 0.0001$), whereas the effects of inter-phenotype competition were not significant in either case ($\alpha_{U,SF} = -0.22$, 95% CI = (-0.57, 0.12), $P = 0.20$; $\alpha_{SF,U} = -0.11$, 95% CI = (-0.25, 0.03), $P = 0.12$; Fig. 3). Unicell growth performance is relatively insensitive to either competitor ($\alpha_{U,U} = -0.37$, 95% CI = (-2.08, 1.33), $P = 0.648$; $\alpha_{U,SF} = -0.46$, 95% CI = (-2.01, 1.08), $P = 0.536$), whereas snowflake growth suffered from competition from both phenotypes ($\alpha_{SF,SF} = -1.25$, 95% CI = (-2.18, -0.32), $P = 0.011$; $\alpha_{SF,U} = -1.20$, 95% CI = (-2.22, -0.18), $P = 0.025$; Supplementary Data 3).

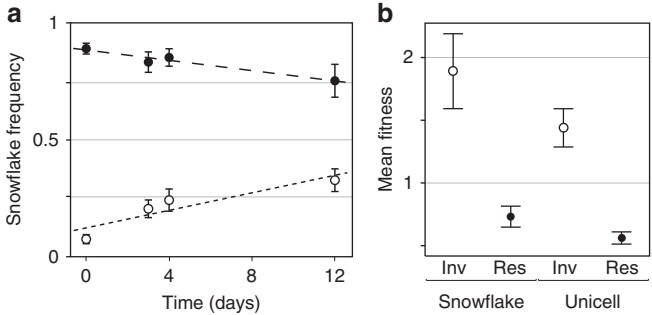

**Figure 2 | Stable coexistence of evolved unicellular and snowflake isolates.** Invasion from rarity experiments were conducted using pairs of snowflake and unicells isolated from each of the ten evolved populations after 60 transfers. (**a**) Time series of mean snowflake (SF) frequencies in populations initiated with snowflakes as invaders (open markers; $n = 10$) and residents (closed markers; $n = 10$). Error bars denote 95% CI. It is noteworthy that data represent SF frequencies in two separate treatments. (**b**) Selection favours invaders across all populations. The fitness of each phenotype is the geometric mean of its relative fitness calculated before and after the final round of settling selection. Markers show the (arithmetic) mean of fitness within both phenotypes as invaders (open markers) and residents (closed markers), and error bars denote the 95% CI. Across all ten populations and two separate mutual invasion experiments, both unicells and snowflakes have significantly higher fitness as invaders than as residents (Tukey's HSD, $n = 40$, $P < 0.0001$; Supplementary Data 3).

**Table 1 | Estimated ecological parameters for both phases of selection are consistent with stable coexistence between unicellular (U) and multicellular *K. lactis* (SF).**

| Selection | $m_{SF}$ | $m_U$ | $\rho$ | $w_{SF}$ | $1/\rho$ | Prediction |
|---|---|---|---|---|---|---|
| Settling | 1.39 | 0.39 | 0.24 | 3.57 | 4.25 | Coexistence |
| Growth | 3.36 | 4.11 | 1.09 | 0.82 | 0.92 | Snowflake excluded |
| Combined | 4.75 | 4.50 | 0.92 | 1.05 | 1.08 | Coexistence |

See Methods for full formulae and Supplementary Data 3 for estimates of all parameters.

The stable persistence of unicellularity depends on two aspects of population responses to settling selection: (i) intense competition among snowflakes, which limits the benefits of cluster formation, and (ii) modest but significant increases in unicell densities. The first aspect is likely to be a result of the fact that the settled material from snowflake-dominated cultures far exceeds the transfer volume; as a result, the probability of survival following successful settling declines with snowflake prevalence. However, the tendency for unicells to increase in density during settling selection is surprising in light of the apparently low settling rates of unicells in monocultures (Fig. 1d), as well as competition from snowflakes for limited transfer volume.

**Sociality strengthens and extends the benefits of multicellularity.** Many wild yeasts can reversibly aggregate into large multicellular structures called flocs, allowing unicells to benefit from increased collective size[34]. Although the ancestral strain *K. lactis* NRL Y-1140 does not visibly flocculate (and is considered nonflocculent), we measured significant flocculation over extended periods (see below). There are multiple ways in which even a modest capacity for social aggregation may influence settling in evolving populations. The efficiency of flocculation increases with particle adhesiveness as well as diameter, and declines with the size disparity between particles[35]. Initially low-flocculation unicellular yeast may therefore increase flocculation through (i) increasing initial particle size by evolving multicellular forms and/ or (ii) increasing cell-level adhesiveness by evolving increased expression (or improved function) of surface lectins involved in flocculation.

In contrast with snowflake clusters, flocs form through social aggregation (CT) of pre-existing cells. Flocs may therefore include unrelated cells, whereas snowflake cluster formation (via ST) excludes non-relatives. Indeed, microscopic observations of

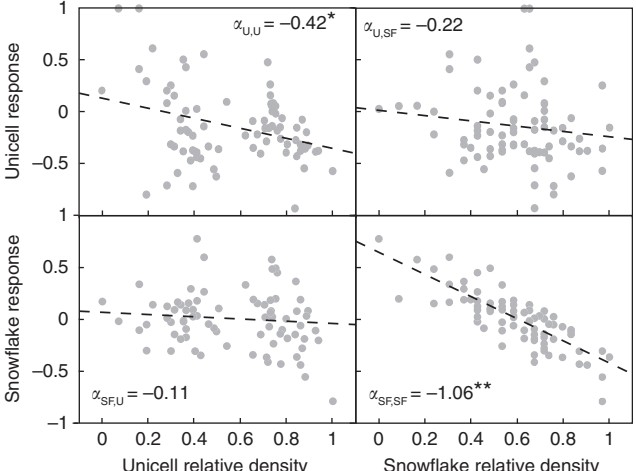

**Figure 3 | Competition during settling is strongest within phenotypes.** Snowflake (SF) and unicell (U) numerical responses to settling (that is, the log ratio of density after and before settling, $m$) have been standardized and centered around zero ($y$ axis) and relative density represents the log density (CFU ml$^{-1}$), scaled from 0 to 1 ($x$ axis). Dashed lines show the prediction from the model response $m_i = c_i + \alpha_{ii} N_i + \alpha_{ij} N_j$, (fit by ordinary least squares) where $c$ is the intercept, $\alpha$ are competition coefficients, $i$ is the focal phenotype, $j$ the opposite phenotype and $N$ is log density of each phenotype before selection. Settling performance declined significantly with increasing abundance of like phenotypes for both unicells ($P = 0.0038$) and snowflakes ($P < 0.0001$), whereas neither cross-phenotype competitive effect was significant (SF effect on unicells: $P = 0.2$, unicell effect on SF: $P = 0.12$; see Supplementary Data 3 for all parameter estimates).

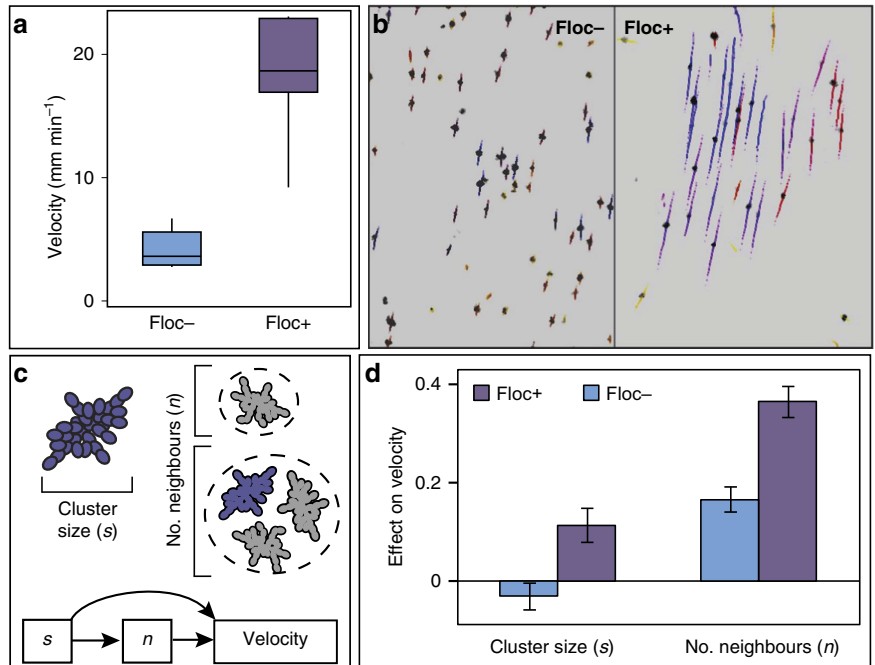

**Figure 4 | Unicellular *K. lactis* benefit by associating with multicellular conspecifics.** (**a**) Unicells (stained red) reversibly adhere to conspecific snowflakes (stained green). (**b**) Unicells that fail to increase when settling alone $m_U = -0.01$, 95% CI ($-0.14$, 0.12)) improve settling performance in mixture with conspecific snowflakes ($m_U = 0.30$, (0.18, 0.43)). However, this benefit disappears when settling with heterospecific snowflakes (*S. cerevisiae*) ($m_U = 0.02$, ($-0.10$, 0.15)). The central line is the median, boxes are inter-quartile ranges and whiskers show the range of measured $m_U$ values. (**c**) Multi-snowflake settling aggregates readily form by indiscriminate attachment among non-relatives during settling. (Pictured are oppositely stained, independently evolved snowflake isolates from Y8 (red) and Y9 (blue) populations.). All scale bars, 25 μm.

**Figure 5 | Flocculation increases *K. lactis* snowflake settling velocity.** (**a**) Disruption of flocculation (Floc −) reduces average snowflake settling velocity relative to settling without floc disruption (Floc +). Central lines show medians, boxes are inter-quartile ranges, and whiskers show the full range of settling velocities observed under each condition. (**b**) Images taken from representative settling videos, in which tracks show snowflake position across 20 frames (=1 s). Track colours correspond to relative velocity. It is noteworthy that only ∼ 5% of snowflake clusters are visible, so individual clusters need not be physically touching to belong to the same floc. (**c**) Path diagram illustrating structural equation model used to estimate the contributions of focal cluster size (diameter) and number of neighbouring clusters (grey) to the settling velocity of a focal cluster (violet). (**d**) The positive effect of neighbours on settling velocity is reduced when flocculation is disrupted. Shown are mean estimates for the contributions of snowflake cluster diameter and number of neighbouring snowflakes to settling in Floc + and Floc − conditions; error bars denote 95% CI.

settled material indicated that unicells adhered to snowflakes in mixed cultures (Fig. 4a), suggesting that aggregates of unicells and multicellular snowflakes may allow unicells to benefit from multicellular neighbours during settling selection. To test this possibility, we measured the effect of snowflakes on the performance of a unicellular strain (Y9U) during standard settling selection. We observed significant differences in settling performance among unicells settling alone and with conspecific and heterospecific snowflakes (ANOVA: F(2,17) = 8.4, P = 0.0036). Unicells did not significantly increase when settling alone ($m_U = -0.01$, 95% CI ($-0.14$, 0.12)), but increased significantly when settling with conspecific snowflakes ($m_U = 0.30$, (0.18, 0.43)). Moreover, unicells co-settled with heterospecific (*S. cerevisiae*) snowflakes experienced no significant

improvement in settling ($m_U = 0.02$, ($-0.10$, 0.15); Fig. 4b), as would be expected if improved unicell settling resulted from proximity to large, fast-settling particles (that is, 'drafting'[36]) or generalized adhesion on the part of unicells. A second set of experiments revealed that diluting *K. lactis* snowflakes to low levels before settling reduced the benefits to unicells (Supplementary Fig. 1 and Supplementary Data 4), consistent with the idea that unicell performance under settling selection depends on snowflake neighbours.

Unicells benefit by transiently associating with multicellular clusters during settling selection (Fig. 4 and Supplementary Fig. 1) without paying the costs of multicellularity (Table 1 and Fig. 1e). However, the capacity to come together with unrelated unicells would seem to be self-defeating, from the snowflake

perspective: unicell adhesion would primarily occur at the periphery of clusters, granting preferential access to nutrients in addition to the benefits of multicellularity. Kin selection frameworks in particular emphasize the threat of exploitation by cheating non-kin as an agent of selection favouring those lineages that avoid non-kin (for example, through unicellular bottlenecks or kin discrimination). Snowflake development necessarily excludes non-kin from clusters via frequent genetic bottlenecks imposed by daughter cells ST following division[25]. Why do genotypes capable of forming clonal groups by ST continue to come together with non-kin, including unicellular free-riders?

This question was addressed by determining the effect of altered flocculation on snowflake settling behaviour. We chemically inhibited flocculation by incubating snowflakes in galactose, which disrupts cell–cell adhesion by binding to carbohydrate recognition domains in the cell surface floc proteins of K. lactis[37]. We then compared the settling behaviour of these low-flocculation snowflakes with those with uninhibited flocculation (incubated in dextrose, which does not influence flocculation) by tracking individual snowflake clusters as they settled in a vertical glass chamber (Supplementary Movies 2 and 3). Highly flocculent snowflakes settled on average fourfold faster $(19.1 \, mm \, min^{-1})$ than low-flocculation snowflakes $(4.1 \, mm \, min^{-1}; \, ANOVA: \, F(1,17) = 88.5; \, P < 0.0001; \, Fig. \, 5a$ and Supplementary Data 5). We next estimated the contributions of focal snowflake cluster size (cluster diameter) and number of neighbouring snowflake clusters (a proxy for overall floc size) to settling speed using a simple structural equation model (Fig. 5c and Supplementary Data 5). Floc size contributed significantly more to settling velocity in high- compared with low-flocculation cultures and significantly exceeded the effect of cluster diameter in both conditions (Fig. 5d).

To validate and extend these microscale results, we measured the effect of high- and low-flocculation conditions on settling in suspensions of multicellular and unicellular (both ancestral and derived) strains. Analyses of time-lapse photographs revealed that flocculation increased settling in the evolved snowflake isolate after 7 m (the duration of settling before selection during the evolution experiment; ANOVA: $F(1,5) = 17.5, \, P = 0.014)$). However, there was no difference between unicell suspensions settled under high- and low-flocculation conditions for 7 m (ancestral unicell: $F(1,5) = 0.71, \, P = 0.447$; evolved unicell: $F(1,5) = 0.11, \, P = 0.760$ Supplementary Data 6). In fact, there was no significant change in the fraction of turbidity after 7 m $(f_{7m})$ in either flocculent $(f_{7m}(ancestor) = 0.998 \, (0.993, \, 1.002);$ $f_{7m}(evolved) = 0.997 \, (0.993, \, 1.002))$ or non-flocculent suspensions $(f_{7m}(ancestor) = 0.996 \, (0.992, \, 1.001); \, f_{7m}(evolved) = 0.996$ $(0.992, \, 1.001))$. Multiple regression analysis revealed a highly significant positive effect of the interaction of snowflake clusters and flocculation on culture settling after 7 min $(P < 0.0001)$. Flocculation did increase the degree of settling in both ancestral and derived unicells after extended settling $(6.5 \, h; \, F(3,35) = 277.1, \, P < 0.0001)$. Interestingly, derived unicells settled significantly faster than ancestors in flocculent suspensions $(P = 0.0036)$, whereas there was no significant difference between ancestral and derived unicells in low-flocculation conditions $(P = 0.51;$ Supplementary Fig. 2 and Supplementary Data 6).

These results suggest that the synergistic interaction of two distinct mechanisms of group formation give rise to a novel collective behaviour in evolving K. lactis populations (Fig. 6a–c). Flocculation is present in both ancestral and derived unicells, but does not significantly influence settling in pure populations of either unicell over a time frame relevant to selection (Fig. 6d). However, flocculation does significantly accelerate settling in multicellular populations. Together, continued association of daughter cells and aggregation of pre-existing (potentially unrelated) cells produce large, multi-cluster flocs during settling (Fig. 5). Although rapid flocculation requires that a fraction of the population invest in permanent multicellularity, the resulting flocs also benefit unicellular 'free-riders' (Fig. 4). Cooperative settling therefore emerges from the interaction of two distinct traits, snowflake cluster formation and flocculation.

**Interspecific differences in settling behaviour.** The evolutionary paths of populations derived from K. lactis and S. cerevisiae sharply diverge following the parallel evolution of snowflake multicellularity: whereas unicells stably persist throughout 60 rounds of settling selection in K. lactis, multicellularity sweeps to fixation in S. cerevisiae[24]. The present work suggests a possible explanation for these different outcomes: K. lactis snowflake clusters come together to form large flocs, ultimately extending the benefits of multicellularity to unicells able to join established flocs. This possibility focuses attention on interactions among snowflake clusters as a potentially important factor in understanding the different evolutionary trajectories of S. cerevisiae and K. lactis populations following the evolution of snowflake multicellularity.

We sought to compare the effect of neighbouring clusters on settling behaviour in K. lactis and S. cerevisiae snowflakes. As both species evolved cluster formation within the same culture conditions (yeast peptone dextrose (YPD) medium) and settling selection protocol, it is possible to compare settling behaviour of these lineages in a common 'native' habitat. We analysed the settling behaviour of individual K. lactis and S. cerevisiae snowflake clusters in low- (5% overnight culture) and high- (100%) density conditions in YPD medium. A 2x2 ANOVA revealed main effects of culture density $(F(1, \, 35) = 39.5;$ $P < 0.0001)$ and the interaction between density and species $(F(1, \, 35) = 23.2; \, P < 0.0001)$, whereas the effect of species was only marginally significant $(F(1, \, 35) = 4.1; \, P = 0.0521)$. Post-hoc analyses using Tukey's HSD showed that the average settling speed of S. cerevisiae snowflakes did not change significantly between low- $(6.8 \, mm \, min^{-1})$ and high-density treatments $(8.5 \, mm \, min^{-1}; \, P = 0.7254)$. In contrast, average K. lactis snowflake settling velocity was four fold higher in dense $(16.3 \, mm \, min^{-1})$ compared with sparse cultures $(3.6 \, mm \, min^{-1}, \, 95\%; \, P < 0.001)$. Furthermore, K. lactis snowflakes settled at nearly twice the rate of the larger S. cerevisiae snowflakes in dense cultures $(P < 0.0002;$ Supplementary Movies 4 and 5), whereas S. cerevisiae snowflakes settled marginally faster than K. lactis in sparse (5%) cultures $(P = 0.2181;$ Fig. 7a,b and Supplementary Data 5).

Local interactions among neighbours had contrasting effects on cluster settling behaviour in K. lactis and S. cerevisiae. Structural equation modeling showed that the local abundance of neighbouring clusters again outweighed focal cluster size as the primary positive influence on settling velocity in K. lactis (mean coefficient value $(c) = 0.53$). In marked contrast, S. cerevisiae snowflake settling was reduced by local crowding $(c = -0.11;$ Fig. 7c), consistent with the observation that competition among neighbours actually hinders settling in dense suspensions of non-flocculent S. cerevisiae snowflakes[36]. Although larger snowflake clusters settled more quickly in both species, this effect was stronger in S. cerevisiae $(c = 0.24)$ compared with K. lactis $(c = 0.10)$. Together, these relationships suggest a fundamental difference between these two species: sedimentation is fastest for large, isolated clusters in the competitively (or solitarily) settling S. cerevisiae, whereas K. lactis settling depends on cooperation among neighbours.

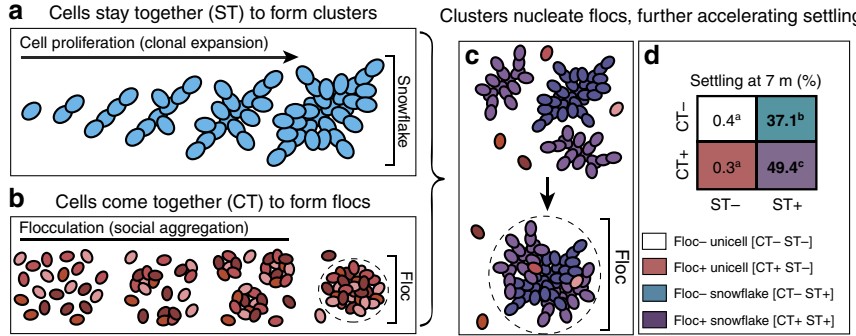

**Figure 6 | Synergy of different paths to multicellularity in _K. lactis_.** Blue, red and violet cells indicate non-flocculent snowflakes, flocculent unicells and flocculent snowflakes, respectively, and different shades denote distinct genotypes within each phenotype. (**a**) Snowflake clusters form by cell proliferation, resulting in discrete groups of permanently attached clonemates. (**b**) Yeast flocs arise through reversible social aggregation, resulting in social groups that may include different genotypes. (**c**) Flocculation is accelerated by multicellular clusters, amplifying and extending benefits of cluster formation. The settling velocity of snowflake clusters increases through floc-mediated cooperation with neighbouring snowflakes; however, cooperative settling may also benefit flocculent unicellular free riders. (**d**) Synergy between cluster and floc formation increases the extent of settling after 7 m in derived _K. lactis_ suspensions. Bold numbers indicate treatments with significant settling (matched pairs _t_-test, _n_ = 3, _P_ < 0.05) and letters denote statistically different groups (Tukey's HSD, _n_ = 12, _P_ < 0.0001) after 7 m. Flocculation significantly increases settling in multicellular populations (_P_ < 0.0001), but does not influence settling in unicellular populations over the duration of standard settling selection (_P_ = 1).

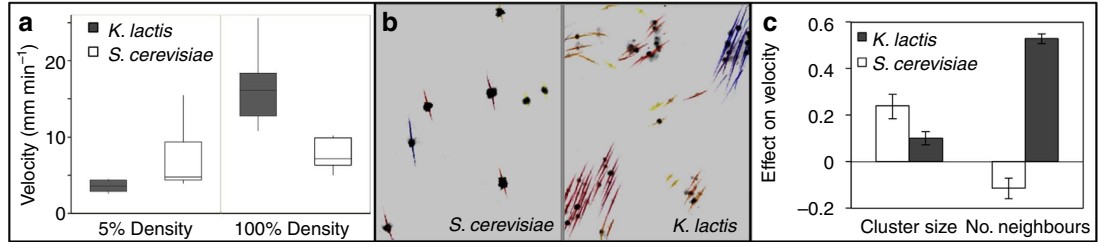

**Figure 7 | Contrasting interactions among multicellular neighbours in _K. lactis_ and _S. cerevisiae_.** (**a**) The settling velocity of _K. lactis_ snowflake clusters increases dramatically in high (100%) compared with low (5%) density conditions (Tukey's HSD: _P_ < 0.0001), whereas _S. cerevisiae_ cluster velocity did not significantly differ between density treatments (_P_ = 0.305). Central lines show medians, boxes are interquartile ranges and whiskers show the full range of observed settling velocities. (**b**) Distinct settling behaviours are evident in settling videos, which show solitary _S. cerevisiae_ snowflakes and cohesive groups of _K. lactis_ snowflakes. Images taken from representative high-density settling videos, in which tracks show snowflake position across 20 frames ( = 1s) and track colours reflect relative cluster velocity. It is noteworthy that only 5% of clusters are stained, so visible clusters need not appear to touch to be physically associated. (**c**) Contrasting interactions among neighbouring snowflakes differentiate competitive and cooperative settling. Focal snowflake cluster size is the primary determinant of settling velocity in multicellular _S. cerevisiae_, whereas neighbouring snowflakes overwhelmingly contribute to rapid settling in _K. lactis_. Neighbouring clusters actually reduce settling velocity in _S. cerevisiae_, consistent with local competition among snowflakes via hindered settling. Error bars denote the 95% CI.

## Discussion

We observed important instances of parallelism and divergence during the experimental evolution of multicellularity in two species of budding yeast in response to a common set of environmental challenges. The evolution of snowflake multi-cellularity occurred reliably in both _S. cerevisiae_ and _K. lactis_, suggesting that this phenomenon is not contingent upon the specific recent evolutionary history of _S. cerevisiae_. However, interactions among multicellular neighbours were strikingly different in these two species: settling in multicellular _S. cerevisiae_ was marked by local competition, whereas _K. lactis_ snowflake clusters transiently assembled into cooperative social groups (Fig. 7b). The benefits of cooperation among multicellular clusters extended to unicellular free-riders (Fig. 4 and Supplementary Fig. 1), promoting the survival of unicellularity in _K. lactis_.

The stable coexistence of multicellular and unicellular genotypes is a notable departure from previous experiments. Genetic variation within multicellular groups promotes cheating and ultimately undermines multicellularity in mat-forming bacteria[23] and slime moulds[14]; conversely, the evolution of clonal clusters appears to preclude cheating in algae[26] as well as in initially flocculent[38] and non-flocculent _S. cerevisiae_[24]. We have found evidence for an intermediate scenario in _K. lactis_: although multicellularity does aid unrelated free-riders (Fig. 4 and Supplementary Fig. 1), snowflake-producing genotypes enjoy preferential access to the benefits of increased size (Table 1). The relative advantage of cluster formation declines sharply with abundance (Fig. 3), giving rise to density-dependent selection for unicellular free riding and promoting stable coexistence (Fig. 2). Density-dependent selection for free riding has been reported for many instances of microbial cooperation in the lab[39–44] and appears prevalent in nature[45–48]. Future work will focus on understanding the genetic and physical factors that limit the access of unicells to the benefits of cooperation in flocculent populations and how these factors influence and respond to subsequent evolution.

Extant multicellular life employs two fundamentally distinct mechanisms of multicellular group formation: continued association of cells following division gives rise to clonal groups (ST), whereas aggregation of pre-existing cells results in social groups that may comprise distinct lineages (CT[12,20]). The continued involvement of both mechanisms sets _K. lactis_ apart from other microbes used to experimentally evolve multicellularity, which

rely exclusively on either ST (bacteria[23], yeast[24], algae[26,28]) or CT (myxobacteria[49] and slime mould[14]). However, dual mechanisms of group formation may be relatively common during various stages of the transition to multicellularity. Many undifferentiated multicellular groups involve both mechanisms (for example, bacterial biofilms[50] and microalgal colonies[51]); furthermore, complex multicellular lineages that rely primarily on ST may possess a surprising capacity for CT, including colonial animals[52] and multicellular red[53], green[54] and brown[55] algae. Although these different paths to multicellularity are often viewed as alternative solutions to similar ecological challenges, these observations suggest that ST and CT may serve different or complementary functions during this major evolutionary transition.

## Methods

**Settling selection experiment.** Ten replicate populations were established with isogenic *K. lactis* strain NRLY-1140 (ATCC 8585; 'Y-1140' or 'ancestor' hereafter). Cultures were grown in 10 ml YPD in 25 mm × 150 mm glass culture vials at 30 °C, shaking at 250 m$^{-1}$. After 24 h, vials were mixed at moderate speed and 1.5 ml aliquots were transferred to 1.5 ml microfuge tubes, which were left undisturbed for 7 min. The bottom 100 µl was then carefully removed with a pipette and transferred to a fresh vial of YPD. Populations were propagated for 60 days. Every tenth transfer, 700 µl samples of each population were placed in 15% glycerol and stored at −80 °C. After 60 days, populations were preserved and streaked onto YPD agar plates to pick individual colonies.

**Transformation.** Strains were transformed with pFA6a-link-yEGFP-KanR[56] (Addgene plasmid 8728) using the lithium acetate/single-stranded DNA/polyethylene glycol (PEG) approach of Geitz et al.[57]. Briefly, *Escherichia coli* carrying plasmid pFA6a-link-yEGFP-KanR were shaken overnight in lysogeny broth (LB) supplemented with ampicillin (100 mg l$^{-1}$) and plasmid DNA was extracted using a PureYield Plasmid Miniprep System (Promega) and diluted to 20 ng µl$^{-1}$ in ddH2O. Yeast cultures were grown up overnight in 10 ml liquid YPD medium and 0.2 ml was transferred into 50 ml liquid 2× YPD medium (40 g dextrose, 40 g peptone, 20 g yeast extract per litre) supplemented with 100 mg adenine per litre. After 8 h of growth, exponentially growing cells were pelleted by centrifugation, aspirated, and resuspended in lithium acetate transformation mix (240 µl 50% polyethylene glycol solution (w/v), 36 µl lithium acetate solution (1.0 M), 50 µl single-stranded DNA from salmon sperm (2.0 mg l$^{-1}$) and 34 µl plasmid DNA per reaction). Cells were incubated in the transformation mix for 20 m at 42 °C, pelleted by centrifuge, aspirated, resuspended in ddH2O and transferred into 10 ml YPD broth for recovery before antibiotic selection. After outgrowth (12 h), cultures were plated onto YPD + G418 (YPD agar supplemented with G418 (200 mg l$^{-1}$)) plates. Transformant colonies were selected after 5–7 days of growth at 30 °C and re-streaked onto fresh YPD + G418 plates to confirm resistance before use in co-settling experiments.

**Population spotting.** Standard plating techniques using spreaders or beads appeared to break multicellular *K. lactis* clusters, preventing reliable determination of CFUs. However, we determined that diluting in ddH2O, vigorous vortexing and spotting (10 µl per spot) on YPD agar plates did not cause breakage of clusters. We therefore used spotting on relatively dry YPD agar plates to enumerate population densities, as well as relative frequencies of different phenotypes. Following spotting, plates were quickly moved to a sterile hood, where the lid was removed and spots were allowed to dry for 10–20 min, to ensure an even distribution of CFUs in the relatively small spots. to avoid difficulties in distinguishing among neighbouring colonies, colonies were counted while still small (after 15–20 h of growth at room temperature on relatively dry plates) using a dissection scope.

**Characterization of populations.** Glycerol stocks were thawed and dilution series were spotted (10 µl per spot) on YPD agar plates, to assess the presence and absence of different phenotypes. Cluster morphology was determined to be a reliable indicator of phenotype by picking several 'smooth' and 'rugose' CFUs, and growing them overnight on YPD. In all cases, 'smooth' populations produced populations dominated by unicells or clusters of two to four cells, which were not readily differentiated from the ancestor, whereas rugose populations were dominated by multicellular 'snowflake' clusters.

Twenty-four hour growth curve data were acquired for a unicellular and snowflake isolate from each of ten populations, as well as the ancestral unicell (strain Y-1140) using a Tecan Infinite 200Pro. Cultures were diluted to 1% of early stationary phase densities in YPD media and four 500 µl replicates were grown for each strain in a 48-well plate. Parameters were estimated by fitting a four-parameter Gompertz model in JMP (JMP Pro, Version 12.1, SAS Institute Inc., Cary, NC, 1989–215) after this model consistently outcompeted logistic (three and four parameters), other Gompertz models (three and five parameters) and Michaelis–Menten models on the basis of akaike information criterion (AIC) for all strains tests.

**Mutual invasion experiments.** Clonal populations of snowflake or unicellular lineages were chosen at random from all ten populations after 60 transfers. Clonal populations were confirmed by re-streaking and growing overnight in YPD. Glycerol stocks were made from these cultures and each phenotype from each population was then inoculated as 10% ('invader') and 90% ('resident') mixtures with the opposite phenotype from the same population (total density 10$^5$ CFU ml$^{-1}$). Populations were immediately spotted, to quantify initial frequencies of both types. Polymorphic populations were then grown under the same settling selection regime as described above. Population state (density of unicell and snowflake CFUs) was determined at various points throughout the experiments by spotting populations immediately before and/or after settling selection. We conducted two independent experiments consisting of one replicate for each treatment (population × initial state), yielding two replicates per population (1–10) per state (snowflake resident and snowflake invader), for a total of 40 populations.

**Quantifying mechanisms of coexistence.** Performance tradeoffs and self-limitation by snowflakes may facilitate coexistence under experimental conditions, which constitute a fluctuating environment. To test this possibility, we estimated the intra- and inter-strain competition coefficients α for both growth and settling selection by least squares regression of enrichment of a focal strain ($m_i$) to density ($N$) of the same ($i$) or opposite ($j$) phenotype (following ref. 32):

$$m_i = m_{i,0} + \alpha_{i,i} N_i + \alpha_{i,j} N_j \tag{1}$$

The degree of niche overlap is then defined as:

$$\rho = \sqrt{\frac{\alpha_{ij}\alpha_{ji}}{\alpha_{ii}\alpha_{jj}}} \tag{2}$$

Finally, fitness similarity ($k_i$) is simply the ratio of the mean absolute fitness of the two competitors:

$$k_i = \frac{\overline{m_i}}{\overline{m_j}} \tag{3}$$

where bars denote arithmetic means. (It is noteworthy that $k_i$ is identical to relative fitness of strain $i$, $w_i$.) Stable coexistence requires that competitors are more similar in average fitness than ecological niche, which is satisfied when $k$ falls between $\rho$ and $1/\rho$.

**Short-term settling experiments.** Overnight cultures of snowflake and/or unicell cultures were combined and adjusted with YPD to a final volume of 1.6 ml, homogenized and a 100 µl sample was immediately removed, diluted and spotted to quantify initial strain densities before settling selection. Cultures were then allowed to settle for 7 m on the bench top and the bottom 100 µl was removed, diluted and spotted following the standard protocol described above. For experiments in which it was not necessary to quantify snowflake CFUs (for example, Supplementary Fig. 1), we made the following modifications on the above protocol: we used a G418-resistant unicell (strain Y9U) and plated mixed populations on antibiotic (G418) YPD agar plates with sterile glass beads. This method was preferable to (time-intensive) spotting, particularly at high snowflake/low unicell densities, because G418-sensitive snowflake colonies cannot grow on G418 plates.

**Snowflake settling video analyses.** Snowflake settling was observed using a customized video microscope apparatus designed by Rebolleda-Gomez et al.[36]. Settling was visualized in both YPD medium (to determine the effects of density on snowflake settling under 'natural' conditions) and ddH2O supplemented with 20 g per litre of glucose or galactose (to determine the effects of flocculation on snowflake settling; ddH2O was used in place of YPD due to the difficulty of blocking flocculation via galactose supplementation in YPD media[58]). Unstained cells were resuspended in the test medium and incubated for 10–30 min with occasionally mixing by vortex. Before observation, 200 µl of overnight cultures were pelleted and stained with 20 µl safranin for ~5 m. Stained cells were then resuspended in ddH2O, pelleted, aspirated and resuspended in 200 µl test medium. Immediately before visualization, 5% (by volume) stained cells were added to unstained cells to yield 200 µl total culture volume, vortexed vigorously for 2–3 s and carefully pipetted into the top of the test medium filling the settling chamber. In some cases, flocculent cultures would accumulate along the edges of the visualization chamber as they settled; careful occasional flexing of the silicone tubing as snowflakes passed this point minimized this issue. Any time that the experimenter directly flexed the tubing, an equilibration time of >1 min was allowed to pass before videos were taken. In both experiments, three technical replicates were conducted per treatment and for each technical replicate, a minimum of three videos with 500 frames (×0.05 s frame−1 = 25 s) were collected and analysed.

Videos were pre-processed and analysed using the TrackMate package[59] in ImageJ. Videos were pre-processed to remove persistent heterogeneities: each video file was inverted and the median value of each pixel (calculated separately for each individual video file) was subtracted from every frame in the video. TrackMate output files were then analysed in JMP Pro. (Representative examples of processed videos with cluster tracks added in TrackMate are available as Supplementary Movies 2–5.) Automated clustering was conducted by first calculating the

nonparametric density profile for each frame of a video, applying modal clustering of the particles, and assigning a local density estimate for each particle in the frame. Particle velocity, size (diameter) and group size were quantified using the mean per frame displacement, median estimated diameter and mean local density, respectively. We used an image of a wire of known thickness to convert displacement and diameter estimates of videos from pixels to mm (M. Rebolleda-Gomez, personal communication). Visual inspection revealed that flocculent, stained *K. lactis* snowflakes were sometimes directly attached, resulting in both clusters being counted as a single particle by TrackMate. (Similar errors could occur when non-flocculent *S. cerevisiae* particles temporarily overlap across a few frames, but these overlaps would not persist and thus did not impact median size estimates.) As multi-snowflake particles are actually social groups (that is, flocs) and we sought to differentiate the effects of snowflake size from social group size, we excluded particles beyond the 97.5 percentile of *K. lactis* strain Y9B snowflake diameters measured by a Multisizer 3 Coulter counter (Beckman Coulter; Supplementary Data 3), which was 33.3 µm (Supplementary Data 3).

**Microscopy.** Fluorescent and bright-field microscopy was conducted using a Nikon A1si confocal microscope. Overnight cultures were diluted in ddH$_2$O and ∼10 µl drops were visualized on standard glass slides with coverslips. A small amount (∼1 ul) of calcofluor white was added to visualize cell wall fluorescence of snowflakes. Images were taken using NIS-Elements AR software (v4.3, Nikon, Tokyo, Japan) and minor processing (brightness and contrast adjustments) was conducted using the platform Fiji[60] in ImageJ[61].

We used light microscopy to visualize co-flocculation between unicells and snowflakes from the same population (Y9), as well as between independently evolved snowflakes from Y8 and Y9 populations. Overnight cultures were pelleted, washed in ddH$_2$O and fixed in 70% ethanol for 5 m. Fixed cells were then washed twice in ddH$_2$O, stained with Congo Red or Methyl Blue for 20 m with occasional mixing and washed twice in ddH$_2$O to remove excess stain. Immediately before visualization, equal volumes (typically 200 µl) of oppositely labelled cells were combined in a 1.5 ml microcentrifuge tube, homogenized vigorously for ∼2 s and allowed to incubate for ∼1 m. Approximately 50–100 µl of settled material was then carefully drawn into a 1 ml pipette tip (modified by clipping ∼2 mm from the tip to allow a wider tip and thus gentler pipetting of fragile aggregates), then 800–850 µl of fresh ddH$_2$O was immediately drawn into the pipette tip below the fixed cell material. The pipette was then held vertically over a chamber slide (containing fresh ddH$_2$O) and settling of mixed culture material through the pipette tip was visually monitored. When the first aggregates approached the pipette tip, the tip was gently swirled through the liquid in the counting chamber to allow settling aggregates to pass from the tip and settle undisturbed into the counting chamber for immediate visualization. Image post-processing was limited to adjusting contrast, color balance, and brightness to maximize image clarity.

**Estimating culture settling with time-lapse photography.** Quantifying the contributions of snowflake cluster formation and flocculation to settling required comparisons among cultures with dramatically different settling rates. To accomplish this, we used time-lapse photography to visualize settling in static cultures derived from ancestral (strain NRLY-1140) and a derived unicellular strain (strain Y9U), as well as a derived snowflake strain (Y9B). Overnight cultures were washed and resuspended in sugar solutions as described above ('Snowflake settling video analyses'). To prevent the formation of disruptive gas bubbles and to minimize the effects of changes in population density or culture conditions over prolonged intervals (6 h), we heat-fixed cultures at 53 °C for 10 min before settling. This treatment was sufficient to kill cells without disrupting flocculation. Cell suspensions were then vigorously homogenized, placed in 12 × 12 × 45 mm polystyrene cuvettes, mixed by pipette and settled for 10 min or 6 h for snowflake and unicellular strains, respectively. Images were acquired once every 15 s for the first 10 m of settling and every 15 m thereafter. Image pre-processing and measurement were conducted using the platform Fiji[60] in ImageJ[61]. Pre-processing was limited to cropping, adjusting image rotation and enhancing contrast. For each replicate and time point, the mean pixel value for fixed area (112 × 379 pixels) within the upper half of the cuvette was calculated. As brightness corresponds to culture turbidity, the extent of settling was quantified as percent reduction from the initial brightness.

**Data availability.** The data that support the findings of this study are available from the corresponding author upon reasonable request.

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

## Acknowledgements

This work was supported by a grant from the John Templeton Foundation. We thank K. Kretman, J. Blum, M. Rebolleda-Gomez and N. Jahren for invaluable assistance in the laboratory, as well as R.F. Denison and W. Harcombe for use of facilities. M. Rebolleda-Gomez designed, shared and assisted with the custom microscope used to visualize cluster settling behaviour. Work was done using Nikon A1 Spectral Confocal Microscope at the University of Minnesota–University Imaging Centers (http://uic.umn.edu). We are indebted to G. Olmedo-Álvarez, L. Eguiarte-Fruns, R.F. Denison and members of the Micropop group for comments on an earlier draft of this manuscript, as well as L.L. Sloat, R. Pollard, W.R. Ratcliff, G. Velicer, W. Harcomb and R. F. Denison for helpful discussions.

## Author contributions

W.W.D. and M.T. conceived of the project, planned and executed the experiments, analysed the data and wrote the paper.

## Additional information

**Competing interests:** The authors declare no competing financial interests.

**Publisher's note**: 

