## [Peer Review File · Nature Communications]

Reviewers' comments:

Reviewer #1 (Remarks to the Author):

This is a clever paper by Driscoll and Travisano that advances our understanding of how multicellularity evolves. This paper builds on prior work where the senior authors' lab artificially evolved multicellular yeast using centrifugation for selection. This results in rapid selection for multicellular yeast that are termed "snowflake" yeast. Significantly, when *S. cerevisiae*, a unicellular organism that does not flocculate, is selected in this manner, it results in snowflakes that reproduce into new snowflakes. I.e. there are no "cheaters". However, in this new work, Driscoll and Travisano repeat the artificial selection experiment on a strain of yeast, *K. lactis*, that has propensity to flocculate depending on growth conditions. When this strain is selected for multicellularity by centrifugation, the authors find that unicellular cells hitch a ride during the centrifugation that makes the snowflakes settle faster, but these unicells are not multicellular snowflakes, but instead reproduce as unicells. Significantly, it appears that the propensity of *K. lactis* to flocculate influences unicellular hitchhikers. The authors then follow this up with mathematical modelling of their fitness and the selective process itself work.

Overall, this is a very well written and potentially nice advance in the field. However, there are some limitations to the work in its present form that limit its potential advances in the field.

Major areas of weakness

1) The primary area of weakness is that the findings related to the "cooperative" flocculation during centrifugation is circumstantial and observational. The molecular basis of flocculation in *K. lactis* is known (and is a genetically heritable trait). While the authors did compare to non-flocculating *S. cerevisiae*, the impact of this work is quite diminished by not exploring the molecular basis of cooperation in more detail. A key limitation comes from the authors' use of a *K. lactis* strain that does not floc, but apparently genetic loci that evolve rapidly may lead to a transition in whether cells flocculate or not. Understanding this more completely is essential to for this paper.

For example, is flocculation/adhesion transiently turned on during selection? Is it constitutively turned on? What is the cost of making the flocculation/adhesion molecule versus strains that do not with, and without artificial selection? What makes this trait more than simply "sticky" cells? The authors do address this with their modeling experiments on the theoretical/predictive side, but given that the molecular tools to address this question exist (e.g. Bellal et al. DOI 10.1016/0032-9592(94)00046-8, El-Behhari et al. DOI 10.1007/s002530051131, Backhaus DOI 10.1002/yea.1781, amongst others), the authors should do this.

In addition to genetic approaches, what about chemical approaches for diminishing the stickiness of cells? Would adding tween-20 reduce the stickiness? What about blocking the cell surfaces with crude cell wall lysates to diminish adhesion? What about treating the cells

with protease (e.g. trypsin) to digest the adhesion molecules? It is possible to control flocculation in *K. lactis* with sugar choice during growth (as the authors appear to have done this with sugar based control of flocculation), but my reading of the literature suggests it is not straight forward to do so.

2) There are several instances where phenotypes are described qualitatively (e.g. Fig 1A-C). These phenotypes should be described quantitatively. E.g. what is the average number of cells per group, what does the distribution look like etc. It would appear the authors are aware of these differences, but having a more complete understanding of what the images in Fig 1A-C represent would be useful.

3) Do these snowflakes undergo programmed cell death as in the case of *S. cerevisiae* snowflakes? Do unicells vs. snowflake cells preferentially undergo programmed cell death? Based on the Ratcliff 2012 results it would suggest that part of the fitness increase could come from reduced cell death with transient adhesion.

Reviewer #2 (Remarks to the Author):

This paper builds on earlier work from Travisano, which showed that selection for rapid settling produced multicellular aggregates that arose because mother and daughter cells failed to separate after cytokinesis. The new features of this work repeating the selection in *Kluyveromyces lactis*, showing that in this organisms the mixtures of multicellular aggregates and single cells exhibit frequency-dependent selection (whoever is in the minority has the selective advantage) and arguing that the *K. lactis* shows the maintenance of two phenotypes because sticky, single cells can settle faster by sticking to clumps and clumps can gain a similar advantage by sticking to each other. The authors clothe these experiments in the provocative title "Synergistic cooperation promotes multicellular performance and unicellular free-rider persistence", but Desai has already published a nice paper showing frequency-dependent selection between two budding yeast populations that have subtly different life styles, which included a mathematical analysis of the phenomenon, (PNAS 112, 11306), the idea and observation that stickiness reduces the selective difference between clumps and single cells seems unremarkable, and I am unconvinced about the arguments about group selection.

Other points

The authors argue they are testing the hypothesis that their previous evolution of multicellularity in *S. cerevisiae* was due to whole genome duplication in this part of the hemiascomycetes by asking whether *K. lactis* can evolve multicellularity. There are two problems with this: 1) the hypothesis is about as idle as speculation can get and 2) whether *K. lactis* can or cannot easily evolve multicellularity has no bearing on the hypothesis, unless the authors are prepared to determine the genetic trajectories that lead to the phenotype in the two organisms.

Phrases like this "Unlike 'directed evolution,' this protocol does not select for specific traits other than fitness; rather, it imposes selection and evolutionary responses are observed. " are profoundly unhelpful: 1) without defining "directed evolution" and citing references, the reader has no idea who is being attacked, 2) many, many studies in experimental evolution select for fitness as faster proliferation, which is no more directed than the selection the authors use, and 3) selecting for how fast cells fall to the bottom of a tube would seem to many to be a somewhat contrived and artificial selection, and the argument that this is a proxy for various forms of more natural natural selection (the duplication is intentional) is unconvincing.

There is a simple experiment to test the idea that stickiness is what leads to the persistence of unicellular cells: varying the density at which cultures are propagated. If the single cells persist in cultures subject to settling selection, their ability to do so will depend on culture density, since collisions between clumps and single cells are required to settle single cells, and the rate of collision will go as the product of the density of the two genotypes. This would give rise to exactly the same sort of mathematical analysis that Desai used to demonstrate that predictions of their model for coexistence were quantitatively met. Another useful experiment would be to control the degree of stickiness in *S. cerevisiae* and show that this could recapitulate the findings in *K. lactis*.

Reviewer #3 (Remarks to the Author):

This manuscript describes the evolution of cooperation among cells of the yeast species *Kluyveromyces lactis*. This species can evolve de novo multicellularity in the lab through a process of incomplete cell division, whereby the daughter cell remains attached to their parent cells, resulting in a 'snowflake' morphology. The morphology readily evolves in response to settling selection, where only individuals that reach the bottom of a tube after a short period of time are transferred to the next generation.

The main results of the manuscript is that, unlike *Saccharomyces cerevisiae*, which also rapidly evolves the snowflake morphology under similar settling selection, in *K. lactis* single cells persist. Invasion from rare assays indicate negative frequency dependence, which suggests a balanced polymorphism between unicellular and multicellular forms. Mechanistically, unicells have shorter lag phases and faster maximal growth rates than snowflakes. Surprisingly they don't have an obvious disadvantage during the settling stage, in that they seem to stick to snowflakes and thus settle significantly better than they do in the absence of snowflakes. The 'free-riding' ability of the unicells seems limited to conspecific pairings. Flocculation seems to improve settling rates in *K. lactis* but not *S. cerevisiae*.

Overall this is a truly interesting, data-rich manuscript that successfully points out the many different levels of cooperation and how different types (unicellular, multicellular) might be maintained through a complex set of interactions. The difference in outcome between *K.*

lactis and *S. cerevisiae* in terms of the maintenance of unicellular forms is an interesting comparison and enriches my understanding of this model system for the evolution of multicellularity.

Comments

1. Despite all of the preceding results that I think are well supported by the data, starting on Page 10, Line 14 to the end, I cannot follow the conclusions. I don't understand the supposed feedbacks, the limited social dilemma (line 17). I think the authors are saying that snowflakes preferentially form flocs, which allows them to settle even faster. Somehow this is considered 'synergy' or synergistic cooperation but I am unclear what that means exactly, how it is formally quantified, and how it differs from just cooperation.

2. I struggled with inconsistent or poorly defined terminology throughout the manuscript. For instance, multicellular clusters<=>snowflakes<=>clonal groups<=>individual clusters –these terms are used interchangeably. In the last sentence of the abstract, it is unclear whether the "social collective" (redundant?) behavior of multicellular clusters is referring to snowflakes/clonal groups/multicellular clusters or aggregates/flocs. I also think of the multicellularity (i.e., snowflake formation) as being a form of cooperation in itself, but the authors at times seem to be reserving the term for the aggregation behavior (flocculation) and elsewhere treating them as different "modes" of cooperation.

Page 10, Line 27 (and throughout) a distinction is made between "clonal" and "social" cooperation. This word choice is strange: clonal refers to a genotypic composition, which may or may not involve cooperation. Cooperation refers to behaviors that produce mutual benefits, and I think is intrinsically social. Elsewhere it is referred to as 'solitary' and 'social'.

Page 8 L5. Unicells are enriched —enriched is typically used in genetic screens to refer to an increase in frequency of one type relative to another. I think the authors mean that unicells are enriched in the sense of becoming concentrated at the bottom of the tube, which may or may not mean enrichment relative to snowflakes.

Page 10 Line 27: synergistic fitness benefits of both modes of cooperation. How is this synergistic and not additive? What does synergistic cooperation mean?

3. An important part of the argument (and included the title) is that unicells are free-riders that gain the benefits of cooperation without paying the costs. This seems to be asserted more than demonstrated from any data. How these unicells constitute free-riders needs to be addressed more carefully.

4. Does any clumping of snowflakes constitute a floc, or is there a more technical definition of flocculation? In addition, differences in levels of flocculation were asserted based on changes to the medium from glucose to galactose. The authors should demonstrate that the manipulation to the medium really does disrupt flocculation as expected, before drawing conclusions about how changes in flocculation affect settling rates.

5. I am confused about how the authors distinguish a large snowflake from a flocculating group. The videos look too blurry to infer potential boundaries between multiple snowflakes. Quantification typically was by plating a sample of the population and looking at colony morphology – but a single snowflake or an aggregate of snowflakes would each produce a single, rugose colony? How does one accurately quantify how many individuals are snowflake vs unicell in populations if they stick to one another?

6. There was a frequent disconnect between the stated results and the statistics that follow in parentheses, which cuts down on clarity and obscures the relationship between the conclusions and the experimental design. For instance, means and CIs are often presented where phrasing implies a t-test/ANOVA between two groups. Some times p-values alone are presented without the test statistic. If the goal is to say a particular value differs from zero, then reporting a CI makes sense (but maybe state one-sample t-test with null hypothesized value of 0), but where the words imply a comparison of two groups (or some other statistical test), the test-statistic, degrees of freedom, and p-value should follow. (See p.4 lines 14-20, p. 9 lines 1-10, p. 9 19-21, p. 19 lines 5-8, p.4 lines 29-30 for examples.) Also, p. 9 lines 19-21 refers to Figure 5F, but there is no Figure 5F.

7. Figure 1E. Do points represent means of multiple growth curves per clonal isolate? Or single estimates per lineage?

Reviewer #1 (Remarks to the Author):

This is a clever paper by Driscoll and Travisano that advances our understanding of how multicellularity evolves. This paper builds on prior work where the senior authors' lab artificially evolved multicellular yeast using centrifugation for selection. This results in rapid selection for multicellular yeast that are termed "snowflake" yeast. Significantly, when *S. cerevisiae*, a unicellular organism that does not flocculate, is selected in this manner, it results in snowflakes that reproduce into new snowflakes. I.e. there are no "cheaters". However, in this new work, Driscoll and Travisano repeat the artificial selection experiment on a strain of yeast, *K. lactis*, that has propensity to flocculate depending on growth conditions. When this strain is selected for multicellularity by centrifugation, the authors find that unicellular cells hitch a ride during the centrifugation that makes the snowflakes settle faster, but these unicells are not multicellular snowflakes, but instead reproduce as unicells. Significantly, it appears that the propensity of *K. lactis* to flocculate influences unicellular hitchhikers. The authors then follow this up with mathematical modelling of their fitness and the selective process itself work. Overall, this is a very well written and potentially nice advance in the field. However, there are some limitations to the work in its present form that limit its potential advances in the field.

Major areas of weakness 1) The primary area of weakness is that the findings related to the "cooperative" flocculation during centrifugation is circumstantial and observational. The molecular basis of flocculation in *K. lactis* is known (and is a genetically heritable trait). While the authors did compare to non-flocculating *S. cerevisiae*, the impact of this work is quite diminished by not exploring the molecular basis of cooperation in more detail. A key limitation comes from the authors' use of a *K. lactis* strain that does not floc, but apparently genetic loci that evolve rapidly may lead to a transition in whether cells flocculate or not. Understanding this more completely is essential to for this paper.

For example, is flocculation/adhesion transiently turned on during selection? Is it constitutively turned on? What is the cost of making the flocculation/adhesion molecule versus strains that do not with, and without artificial selection?

We completely agree that understanding regulation and variation in different aspects of the flocculation phenotype would be very interesting (in fact, we are in the early stages of a transcriptome project intended to illuminate such aspects of derived strains). However, constructing non-flocculent mutants would require an understanding of the molecular basis of this phenotype that is simply beyond the field at the present. The reviewer has pointed to studies that have advanced knowledge of flocculation in K. lactis, and we have performed additional experiments intended to meet the standard set by these studies (see below). Nevertheless, we have been unable to find any studies that provide the kinds of tools that would be required for genetically manipulating flocculation or directly measuring expression of underlying genes.

More broadly, flocculation is a highly complex, multi-locus phenomenon that remains a major challenge even in the model system S. cerevisiae (two recent reviews: doi:10.1111/j.1365-2672.2010.04897.x, doi:10.1111/j.1574-6976.2011.00275.x). Far less is known about molecular aspects of floc in other yeasts, including Kluyveromyces.

We have therefore focused on improving and extending our quantification of floc in the present study by conducting several additional experiments and clarifying our discussion of flocculation in the manuscript.

What makes this trait more than simply “sticky” cells?

*Cells with generalized “stickiness” would adhere to other particles indiscriminately; however, the failure of unicells to increase in density through co-settling with *S. cerevisiae* snowflakes (Figure 5 in the revised document) argues against this possibility. Furthermore, the fact that adhesion is chemically blocked by galactose and melibiose (but not glucose) provides a direct link to flocculation specifically (see below).*

The authors do address this with their modeling experiments on the theoretical/predictive side, but given that the molecular tools to address this question exist (e.g. Bellal et al. DOI 10.1016/0032-9592(94)00046-8, El-Behhari et al. DOI 10.1007/s002530051131, Backhaus DOI 10.1002/yea.1781, amongst others), the authors should do this.

We have conducted additional experiments along the lines of the references provided, which we believe have strengthened the paper (starting Page 7, Line 17; see Figure 7 and Figure S1).

In addition to genetic approaches, what about chemical approaches for diminishing the stickiness of cells? Would adding tween-20 reduce the stickiness?

This is an interesting idea, but we were concerned about additives that may alter settling by changing the fluid viscosity.

What about blocking the cell surfaces with crude cell wall lysates to diminish adhesion? What about treating the cells with protease (e.g. trypsin) to digest the adhesion molecules? It is possible to control flocculation in *K. lactis* with sugar choice during growth (as the authors appear to have done this with sugar based control of flocculation), but my reading of the literature suggests it is not straight forward to do so.

*We have clarified our original methods and conducted additional experiments in which sugars (mainly galactose) are used to chemically disrupt flocculation in cultures killed by heating to 55C for 15 minutes. We used non-viable cells in these experiments to eliminate the possibility of differential regulation of floc gene expression in response to different sugars, i.e. to ensure that the observed changes were due to chemical disruption of flocculation, rather than dynamic expression of floc lectins. Galactose disrupts flocculation in *K. lactis* by binding to the carbohydrate-binding domains within surface lectins, and is therefore an ideal means of reducing flocculation with minimal changes to the physical properties of the medium (e.g. viscosity changes due to addition of surfactants) or of the cells themselves (see below).*

We nevertheless attempted other approaches to disrupting flocculation as suggested by the reviewer, with mixed results. Excessive heating (80C) denatures floc lectins, according to Bellal et al, Proc. Biochem (1995), and others. We found that the resulting suspensions settled remarkably quickly (to our initial surprise); however, subsequent microscopy revealed major changes to cell and colony morphology compared with the more ‘gentle’ 55C treatment.

Snowflakes exposed to 80C had taken on a very round, nearly spherical shape, and constituent cells had shrunk substantially. This is problematic for many reasons, including the possibility that excessive heating may influence the buoyancy of cells. Finally, on multiple occasions we were unable to observe any evidence for disrupted flocculation in either of two proteases (pronase and pepsin).

Bellal et al. found that the effect of different proteases on flocculation in K. lactis varies across strains, whereas sugars have remarkably consistent effects across these same strains. Those authors suggest that this disparity may reflect the existence of conserved carbohydrate-recognition domains within flocculins, whereas other domains within flocculins may vary substantially, rendering individual strains more or less sensitive to particular proteases. We consider galactose-sensitivity necessary and sufficient to demonstrate that flocculation is responsible for aggregation among non-viable K. lactis cells/clusters.

Furthermore, we believe that the new experiments included in the revised manuscript address the central issue—whether flocculation (rather than ‘generalized stickiness’) is the mechanism behind collective settling. We have used galactose (as well as melibiose) to disrupt flocculation in both live and dead cultures, including early (10th transfer) and late (60th transfer) snowflake isolates as well as ancestral and derived unicells.

2) There are several instances where phenotypes are described qualitatively (e.g. Fig 1A-C). These phenotypes should be described quantitatively. E.g. what is the average number of cells per group, what does the distribution look like etc. It would appear the authors are aware of these differences, but having a more complete understanding of what the images in Fig 1A-C represent would be useful.

We have added a supplementary table of summary statistics that quantitatively describe the shapes of individual cells as well as ‘colony-forming units’ (multicellular clusters in the case of snowflakes; single cells or small clusters of 2-4 cells in unicells) for snowflake isolates from all ten populations, as well as the ancestral unicell and a representative derived unicell.

3) Do these snowflakes undergo programmed cell death as in the case of *S. cerevisiae* snowflakes? Do unicells vs. snowflake cells preferentially undergo programmed cell death? Based on the Ratcliff 2012 results it would suggest that part of the fitness increase could come from reduced cell death with transient adhesion.

We have encountered difficulties in addressing this reasonable question using the dihydrorhodamine 123 method of Ratcliff et al., PNAS, 2012. These stem from the presence of a (presently unknown) source of autofluorescence within yeast snowflakes. We have found that cells within snowflakes are propidium iodide-positive; however, we cannot at this point say that these dead cells are evidence of apoptosis specifically.

Reviewer #2 (Remarks to the Author): This paper builds on earlier work from Travisano, which showed that selection for rapid settling produced multicellular aggregates that arose because mother and daughter cells failed to separate after cytokinesis. The new features of this work repeating the selection in *K. lactis*, showing that in this organisms the mixtures of multicellular aggregates and single cells exhibit

frequency-dependent selection (whoever is in the minority has the selective advantage) and arguing that the *K. lactis* shows the maintenance of two phenotypes because sticky, single cells can settle faster by sticking to clumps and clumps can gain a similar advantage by sticking to each other. The authors clothe these experiments in the provocative title "...", but Desai has already published a nice paper showing frequency-dependent selection between two budding yeast populations that have subtly different life styles, which included a mathematical analysis of the phenomenon, (PNAS 112, 11306), the idea and observation that stickiness reduces the selective difference between clumps and single cells seems unremarkable, and I am unconvinced about the arguments about group s.

*We have extensively revised and expanded on our previous explanation of the broader context and significance of our results in the hopes of addressing some of these criticisms. We agree that the demonstration of frequency-dependence in yeast is, by itself, not particularly novel. We have tried to emphasize the truly novel aspects of this work—that interactions with non-relatives generate a (limited) niche for unicellular ‘free-riders’ not observed during the parallel evolution of snowflake multicellularity in *S. cerevisiae*. The field has been historically quite focused on ‘cheaters’ that exploit and ultimately undermine multicellularity (as in Hammerschmidt et al., Nature, 2014), and the observation that cooperation with non-kin persists despite free riders contradicts this conventional wisdom.*

*Finally, we agree that there are some interesting parallels between the present results and the paper from Desai’s group (Frenkel et al, 2015, PNAS). However, our manuscript fills an important gap currently between the two (largely binary) types of outcome observed in past evolution experiments concerning the evolution of multicellularity: (i) the total collapse of cooperation (as observed in the *Pseudomonas fluorescens* SBW25 system, e.g. Hammerschmidt et al.) and the total absence of cheating/free-riding due to complete restriction of cooperation to clonal groups (as observed in *S. cerevisiae* (Ratcliff et al., 2012, PNAS) and *Chlamydomonas reinhardtii* (Ratcliff et al., 2013, Nat. Comm.)). Furthermore, stable coexistence in *K. lactis* populations appears to stem from an interaction between two ‘classes’ of group formation (i.e. ‘coming together’ and ‘staying together,’ Tarnita et al., 2013, J. Theor. Biol.), which are often viewed as strict alternatives and seldom considered together.*

Other points The authors argue they are testing the hypothesis that their previous evolution of multicellularity in *S. cerevisiae* was due to whole genome duplication in this part of the hemiascomycetes by asking whether *K. lactis* can evolve multicellularity. There are two problems with this: 1) the hypothesis is about as idle as speculation can get and 2) whether *K. lactis* can or cannot easily evolve multicellularity has no bearing on the hypothesis, unless the authors are prepared to determine the genetic trajectories that lead to the phenotype in the two organisms.

*We cannot claim credit for this hypothesis, which was advanced by Out et al. 2013 (PNAS). We have revised this section in order to more accurately reflect the motivation of the current work, which stemmed from longstanding questions about contingency in evolution rather than a desire to test this rather limited hypothesis about *S. cerevisiae*.*

Nevertheless, we feel that it is important to (at a minimum) note that the current results are inconsistent with the view that the results of Ratcliff et al. (2012) depended upon the specific

recent evolutionary history of Saccharomyces. (Such assertions include those based on the genome duplication event as well as the claim that unicellular S. cerevisiae are derived from a recent multicellular ancestor.)

Phrases like this "Unlike 'directed evolution,' this protocol does not select for specific traits other than fitness; rather, it imposes selection and evolutionary responses are observed. " are profoundly unhelpful: 1) without defining "directed evolution" and citing references, the reader has no idea who is being attacked, 2) many, many studies in experimental evolution select for fitness as faster proliferation, which is no more directed than the selection the authors use, and 3) selecting for how fast cells fall to the bottom of a tube would seem to many to be a somewhat contrived and artificial selection, and the argument that this is a proxy for various forms of more natural natural selection (the duplication is intentional) is unconvincing.

We have removed this clumsily-phrased contrast with directed evolution, which we did not intend as a slight. We acknowledge that settling selection is artificial, but argue that it still captures key aspects of selection against unicells in nature. (For instance, size-limited predators promote the evolution of undifferentiated multicellularity in unicellular algae, both in the laboratory (Becks et al., 2012, Ecol Lett) and in nature. How artificial is 'too' artificial is ultimately determined on the questions being asked, and we believe settling selection is justifiable from the perspective of the questions we address here.

There is a simple experiment to test the idea that stickiness is what leads to the persistence of unicellular cells: varying the density at which cultures are propagated. If the single cells persist in cultures subject to settling selection, their ability to do so will depend on culture density, since collisions between clumps and single cells are required to settle single cells, and the rate of collision will go as the product of the density of the two genotypes. This would give rise to exactly the same sort of mathematical analysis that Desai used to demonstrate that predictions of their model for coexistence were quantitatively met. Another useful experiment would be to control the degree of stickiness in *S. cerevisiae* and show that this could recapitulate the findings in *K. lactis*.

The work of Desai et al is interesting, but distinct from our current study. Diluting prior to settling is a solid idea, but influences more than the unicell benefit from co-settling: flocculation also promotes coexistence by increasing (global) competition among snowflakes, by (i) increasing overall settling rates, and (ii) increasing the volume occupied by each snowflake. (Floc-mediated adhesion among snowflakes effectively slows the rate at which settled material approaches a final, minimum volume.) Considering the snowflake settling advantage is greatest as low snowflake density, pre-settling dilution could eliminate or reduce unicells through either (i) relaxed intra-phenotype competition among snowflakes, and/or (ii) reduced inter-phenotype facilitation of unicells.

We hope that future analyses of the transcriptomes of derived lines will reveal target loci for floc knockouts, which would allow us to directly control the degree of flocculation in unicells and snowflakes independently.

Reviewer #3 (Remarks to the Author): This manuscript describes the evolution of cooperation among cells of the yeast species *Kluyveromyces lactis*. This species can evolve de novo multicellularity in the lab through a process of incomplete cell division, whereby the daughter cell remains attached to their parent cells, resulting in a 'snowflake' morphology. The morphology readily evolves in response to settling selection, where only individuals that reach the bottom of a tube after a short period of time are transferred to the next generation.

The main results of the manuscript is that, unlike *Saccharomyces cerevisiae*, which also rapidly evolves the snowflake morphology under similar settling selection, in *K. lactis* single cells persist. Invasion from rare assays indicate negative frequency dependence, which suggests a balanced polymorphism between unicellular and multicellular forms. Mechanistically, unicells have shorter lag phases and faster maximal growth rates than snowflakes. Surprisingly they don't have an obvious disadvantage during the settling stage, in that they seem to stick to snowflakes and thus settle significantly better than they do in the absence of snowflakes. The 'free-riding' ability of the unicells seems limited to conspecific pairings. Flocculation seems to improve settling rates in *K. lactis* but not *S. cerevisiae*.

Overall this is a truly interesting, data-rich manuscript that successfully points out the many different levels of cooperation and how different types (unicellular, multicellular) might be maintained through a complex set of interactions. The difference in outcome between *K. lactis* and *S. cerevisiae* in terms of the maintenance of unicellular forms is an interesting comparison and enriches my understanding of this model system for the evolution of multicellularity.

Comments 1. Despite all of the preceding results that I think are well supported by the data, starting on Page 10, Line 14 to the end, I cannot follow the conclusions. I don't understand the supposed feedbacks, the limited social dilemma (line 17). I think the authors are saying that snowflakes preferentially form flocs, which allows them to settle even faster. Somehow this is considered 'synergy' or synergistic cooperation but I am unclear what that means exactly, how it is formally quantified, and how it differs from just cooperation.

We have essentially re-written this entire part of the manuscript, as we agreed with the reviewer's evaluation. We have also revised and expanded the crucial section of the manuscript that introduces flocculation and its significance (additions start on Page 7, Line 17) and conducted additional experiments to more directly illustrate the synergy between the two forms of cooperation (snowflake cluster-formation, and flocculation; new Figure 7 and S1) and dedicated more time to clearly explaining the evidence and significance of this synergy.

2. I struggled with inconsistent or poorly defined terminology throughout the manuscript. For instance, multicellular clusters<=>snowflakes<=>clonal groups<=>individual clusters –these terms are used interchangeably. In the last sentence of the abstract, it is unclear whether the "social collective" (redundant?) behavior of multicellular clusters is referring to snowflakes/clonal groups/multicellular clusters or aggregates/flocs. I also think of the multicellularity (i.e., snowflake formation) as being a form of cooperation in itself, but the authors at times seem to be

reserving the term for the aggregation behavior (flocculation) and elsewhere treating them as different “modes” of cooperation.

Another fair point. We have focused on clarity in our revision, and paid particular attention to semantic consistency and avoiding redundant terms. We have also added a new figure (Figure 7), which we hope will clarify the different types of cooperation, as well as the relationship between the two.

Page 10, Line 27 (and throughout) a distinction is made between “clonal” and “social” cooperation. This word choice is strange: clonal refers to a genotypic composition, which may or many not involve cooperation. Cooperation refers to behaviors that produce mutual benefits, and I think is intrinsically social. Elsewhere it is referred to as ‘solitary’ and ‘social’.

We have adopted the terminology of Tarnita et al., 2013 (J. Theor. Biol) and now refer to cluster-formation as ‘staying together’ and flocculation as ‘coming together’. (We note similar distinctions have been made for some time, at least since Queller’s distinction between ‘fraternal’ and ‘egalitarian’ transitions.)

Page 8 L5. Unicells are enriched —enriched is typically used in genetic screens to refer to an increase in frequency of one type relative to another. I think the authors mean that unicells are enriched in the sense of becoming concentrated at the bottom of the tube, which may or may not mean enrichment relative to snowflakes.

This is an important point to keep as clear as possible, so we have revised these passages to emphasize that it is unicell densities that are increased through co-settling whenever we discuss these results.

Page 10 Line 27: synergistic fitness benefits of both modes of cooperation. How is this synergistic and not additive? What does synergistic cooperation mean?

*We have conducted additional experiments designed to quantify the impacts of flocculation, multicellularity, and the interaction between them on settling rates over short (7 minutes) and longer time frames (new Figures 4 and S1, Table S4). These experiments show that flocculation does not influence unicell settling over durations relevant to selection (7 minutes). In fact, unicells do not settle this quickly under **any** conditions. In our populations, multicellularity appears to be necessary to accelerate flocculation sufficiently to impact settling, yielding a non-additive (synergistic) interaction.*

3. An important part of the argument (and included the title) is that unicells are free-riders that gain the benefits of cooperation without paying the costs. This seems to be asserted more than demonstrated from any data. How these unicells constitute free-riders needs to be addressed more carefully.

We have revised the passages that report and interpret the results of the mutual invasion from rarity experiments (Table 1 and Figure 3) as well as the short-term co-settling experiments (now Figure 5), to make this point more clearly. Based on the analyses reported in Table 1, unicell persistence hinges on their ability to increase during settling selection in mixed cultures. However,

we know that unicells cannot increase in density when settling alone (Figure 5), nor with S. cerevisiae snowflakes. The density increases of unicells when co-settled with K. lactis snowflakes (i.e. the benefit of free-riding) therefore provides an explanation for the tendency of unicells to increase during settling selection in the mutual invasion experiments. We argue this is a case of free-riding in light of the costs of multicellularity (i.e. the fitness difference between unicells and snowflake lineages in Table 1).

4. Does any clumping of snowflakes constitute a floc, or is there a more technical definition of flocculation? In addition, differences in levels of flocculation were asserted based on changes to the medium from glucose to galactose. The authors should demonstrate that the manipulation to the medium really does disrupt flocculation as expected, before drawing conclusions about how changes in flocculation affect settling rates.

As we explained in our response to reviewer 1, we have tried several alternative routes to chemically disrupting flocculation. Of these, we believe the most convincing new data come from the differential effects of galactose, melibiose, and glucose on flocculation in cells that have been heat-fixed at (55C). (Higher temperatures noticeably disrupted cell- and cluster-level morphology, unfortunately.) These effects are evident in ancestral and derived unicells, as well as evolved snowflakes.

Galactose and melibiose are well known to disrupt flocculation in K. lactis lineages by binding to (and thus blocking) the carbohydrate recognition domains in floc lectins at the cell surface. Although strains differ in sensitivity to other approaches to disrupting flocculation (including temperature and different proteases), Bellal et al. 1995 (doi:10.1016/0032-9592(94)00046-8) found that the effects of sugars were remarkably consistent. Galactose is reliably effective at disrupting flocculation, so we mostly focused on this sugar. We also used melibiose, which also significantly slowed snowflake settling (Table S3.) Again, these most recent experiments have used non-viable cultures in order to bypass the potential for dynamic floc regulation or other indirect effects, and therefore to isolate the direct effects of these sugars on floc lectin-mediated cell aggregation.

5. I am confused about how the authors distinguish a large snowflake from a flocculating group. The videos look too blurry to infer potential boundaries between multiple snowflakes. Quantification typically was by plating a sample of the population and looking at colony morphology – but a single snowflake or an aggregate of snowflakes would each produce a single, rugose colony? How does one accurately quantify how many individuals are snowflake vs unicell in populations if they stick to one another?

Only a small fraction (5%) of the dense snowflake cultures used in the settling videos were stained in order to minimize physical contact between stained cells. (We note this point in the captions of Figures 5 and 7, as unstained clusters may actually occupy apparently blank spaces. Clusters that appear separated may therefore occupy the same floc.) Nevertheless, the tracking software did occasionally register multiple clusters as a single cluster. We identified these instances by comparing the distributions of cluster sizes from videos with those obtained from measuring dilute cultures of the same strain with a particle counter (Beckman-Coulter Multisizer 3). In

addition to several other quality control measures (explained more fully in the Methods section), we excluded clusters beyond the 97.5th percentile of diameters as estimated by the particle counter (those over 33.3 μ m) from structural equation modeling.

We have found that the flocs that form during K. lactis settling are readily disintegrated by vigorous vortexing at each step of a serial dilution. (Floc re-formation becomes less likely as density declines.) We have taken several steps to check for systematic biases imposed by flocculation: 'de-flocculating' cultures by performing serial dilutions in galactose (rather than distilled water) does not significantly change counts, and re-streaking single snowflake colonies from mixed cultures has never yielded smooth (unicellular) colonies. We discovered early on that it can be difficult to distinguish among proximate rugose snowflake colonies if they have grown too large. We avoided this problem by using dry plates for our counts and growing colonies at room temperature for < 20 hours. (We have added this important detail to the revised Methods section.)

6. There was a frequent disconnect between the stated results and the statistics that follow in parentheses, which cuts down on clarity and obscures the relationship between the conclusions and the experimental design. For instance, means and CIs are often presented where phrasing implies a t-test/ANOVA between two groups. Some times p-values alone are presented without the test statistic. If the goal is to say a particular value differs from zero, then reporting a CI makes sense (but maybe state one-sample t-test with null hypothesized value of 0), but where the words imply a comparison of two groups (or some other statistical test), the test-statistic, degrees of freedom, and p-value should follow. (See p.4 lines 14-20, p. 9 lines 1-10, p. 9 19-21, p. 19 lines 5-8, p.4 lines 29-30 for examples.) Also, p. 9 lines 19-21 refers to Figure 5F, but there is no Figure 5F.

We thank the reviewer for noticing these inconsistencies, and have corrected them as suggested throughout the text.

7. Figure 1E. Do points represent means of multiple growth curves per clonal isolate? Or single estimates per lineage?

Each point is a mean of four different replicates; we have clarified this in the Figure 1 legend.

Reviewers' comments:

Reviewer #1 (Remarks to the Author):

As described in my previous review, this is an interesting paper that is now further strengthened by addressing my concerns in the prior review with new experiments. In particular, the additional experiments regarding "stickiness" vs. "cooperative flocculation" significantly improve the impact of the results. This was also a highlight weakness by reviewer #3.

However, prior to publication, the authors should consider the following set of revision. The manuscript in its current state is a bit challenging to read and understand the rationale. The introduction in particular needs some attention for clarity. The following suggestions for revision would significantly improve the quality and impact of the manuscript.

In particular, the authors could address a concern of reviewer #2 ("Other points The authors argue they are testing the hypothesis that their previous evolution of multicellularity in *S. cerevisiae* was due to whole genome duplication in this part of the hemiascomycetes by asking whether *K. lactis* can evolve multicellularity. There are two problems with this: 1) the hypothesis is about as idle as speculation can get and 2) whether *K. lactis* can or cannot easily evolve multicellularity has no bearing on the hypothesis, unless the authors are prepared to determine the genetic trajectories that lead to the phenotype in the two organisms.") simply by revising the introduction and rationale for performing selection on *K. lactis*. The paragraph in the intro setting up the paper as it currently does makes it difficult to read and understand.

Otherwise, this is a great paper with just a few rough edges that need to be worked out before publication.

Text revisions for clarity

Line 35-36, this part of the sentence should be deleted to keep the intro on point :

as exemplified by cancer², social parasitism³, and meiotic drive⁴

Line 36, delete "interdisciplinary"

Line 40, revise "According to this theory..." to "According to kin selection, the cost of cooperation can be offset by benefits to relatives, which are more likely to share alleles associated with cooperation".

Line 44-45, revise to read "The origin and maintenance of cooperation among formerly free-living unicells is often viewed as a barrier for the evolution of complex multicellular life^{1,9,10}"

Line 47, revise to read "suggesting that solutions to evolving cooperation have evolved multiple independent times¹¹"

Line 52-54, revise to read "Nevertheless, cooperation among non-relatives has been documented multiple times^{16,17}, including altruism during the development of multicellular chimeras¹⁸."

Line 54-57, this sentence is confusing are the authors referring to selection or that these cells are "selective" in who they cooperate with. It is at this point that the authors introduce a new term for the manuscript "selectivity". I think selectivity makes this part hard to read. Revise to read "These observations suggest that other factors can limit the advantages of genetic similarity for cooperation, though measuring the cost and benefits of cooperativity is challenging."

The introduction should have a brief introduction to sociality regarding multicellularity. This is a critical oversight. There is discussion of kin selection, but a brief discussion of clonal vs. social multicellular mechanisms would be useful (3-4 sentences). E.g. summarize and refer to Grosberg 2007.

A brief introduction to *K. lactis* and its floc would be useful.

Lines 65-75 needs revision. There are a lot of ideas packed into this paragraph that are not fully completed.

Lines 78-81 need revision. The set up for the series of experiments in these two sentences make the paragraph hard to read. My suggestions is "The goal of this work is to understand the outcome selection by sedimentation on *K. lactis*, an organism with conditional cooperative flocculation, compared to the outcome of *S. cerevisiae* which is not flocculant. Factors such as could also influence the outcome of the experiment".

As this paragraph currently reads, the paper appears to be much less interesting than it actually is. It is not until line ~180 that the idea of *K. lactis* being flocculant versus non-flocculant is introduced. I would frame the manuscript in terms of the interesting bit. The rest of the notable differences to *S. cerevisiae* and *K. lactis* noted here are a bit of red herring. The experimental test that is being performed here is *K. lactis* floc vs. non-floc. Reviewer #2 criticizes the paper for focusing on these as possible reasons for differences between strains. This paragraph really needs to be revised to address reviewer #2's concerns as well as make the manuscript more understandable as to what is really being tested here.

Line 202, there is a stray period.

Lines 215-217, change ", which" to "that". The sentence itself should be revised for clarity.

Lines 217-222, revise to read "Snowflake development necessarily excludes non-kin from clusters via frequent genetic bottlenecks imposed by daughter cells staying together

following division³⁴. Why does aggregation with non-kin (including possible cheaters) persist in lineages that already form multicellular clusters with close kin?"

Line 223, revise to read "This question was addressed by altering flocculation and determine the effect on settling behavior".

Reviewer #2 (Remarks to the Author):

Travisano ReReview 2016

This is clearly going to be an editorial decision. The glass half full view is that the authors have made a detailed response to every point made by the reviewers, agree with many of the points they raise, clarified their writing, and performed new experiments. The glass half empty view is that in many cases the detailed response is an argument for not doing an experiment, and in several cases, such as trying to disrupt flocs while they are alive, the experiment is one that has been suggested by multiple reviewers.

Responses to points I raised

*"This paper builds on earlier work from Travisano, which showed that selection for rapid settling produced multicellular aggregates that arose because mother and daughter cells failed to separate after cytokinesis. The new features of this work repeating the selection in *Kluyveromyces lactis*, showing that in this organisms the mixtures of multicellular aggregates and single cells exhibit frequency-dependent selection (whoever is in the minority has the selective advantage) and arguing that the *K. lactis* shows the maintenance of two phenotypes because sticky, single cells can settle faster by sticking to clumps and clumps can gain a similar advantage by sticking to each other. The authors clothe these experiments in the provocative title "Synergistic cooperation promotes multicellular performance and unicellular free-rider persistence", but Desai has already published a nice paper showing frequency-dependent selection between two budding yeast populations that have subtly different life styles, which included a mathematical analysis of the phenomenon, (PNAS 112, 11306), the idea and observation that stickiness reduces the selective difference between clumps and single cells seems unremarkable, and I am unconvinced about the arguments about group selection."*

The authors argue that their work makes two fundamental contributions: 1) it shows cooperative interactions between non-relatives, 2) it allows the selection of cheats that do not destroy the original cooperation. My concern is that both of these phenomena are specialized consequences of the particular selection for faster sedimentation, and that neither result would be generalizable to more traditional forms of cooperation, such as the collective use of public goods. To me this is a reflection of the overemphasis on "levels of selection" and exception to rules that excites a certain section of evolutionary biologists.

*The authors argue they are testing the hypothesis that their previous evolution of multicellularity in *S. cerevisiae* was due to whole genome duplication in this part of the*

hemiascomycetes by asking whether K. lactis can evolve multicellularity. There are two problems with this: 1) the hypothesis is about as idle as speculation can get and 2) whether K. lactis can or cannot easily evolve multicellularity has no bearing on the hypothesis, unless the authors are prepared to determine the genetic trajectories that lead to the phenotype in the two organisms.

The authors' response is satisfactory.

Phrases like this "Unlike 'directed evolution,' this protocol does not select for specific traits other than fitness; rather, it imposes selection and evolutionary responses are observed. " are profoundly unhelpful: 1) without defining "directed evolution" and citing references, the reader has no idea who is being attacked, 2) many, many studies in experimental evolution select for fitness as faster proliferation, which is no more directed than the selection the authors use, and 3) selecting for how fast cells fall to the bottom of a tube would seem to many to be a somewhat contrived and artificial selection, and the argument that this is a proxy for various forms of more natural natural selection (the duplication is intentional) is unconvincing.

The authors' address the first two points and argue that "How artificial is 'too' artificial is ultimately determined on the questions being asked, and we believe settling selection is justifiable from the perspective of the questions we address here." The difference between their position and mine is a reflection of fundamental differences about how, in evolutionary questions, one should argue from the concrete to the abstract and cannot be easily resolved.

There is a simple experiment to test the idea that stickiness is what leads to the persistence of unicellular cells: varying the density at which cultures are propagated. If the single cells persist in cultures subject to settling selection, their ability to do so will depend on culture density, since collisions between clumps and single cells are required to settle single cells, and the rate of collision will go as the product of the density of the two genotypes. This would give rise to exactly the same sort of mathematical analysis that Desai used to demonstrate that predictions of their model for coexistence were quantitatively met. Another useful experiment would be to control the degree of stickiness in S. cerevisiae and show that this could recapitulate the findings in K. lactis.

The authors ignore the second suggestion and argue against the first, claiming that dilution will affect more than the collision between single cells and snowflakes. My position is that as long as the experiment is performed under conditions where snowflakes settle, whether they stick to each other or not, the effect of dilution should be to reduce the benefit to single cells, and that showing this to be true would go a long way to support the model.

Reviewer #3 (Remarks to the Author):

Overall I am satisfied with the revisions, and most of my questions have been answered. I

like the “coming together” or “staying together” vocabulary, and I think it is a very useful distinction (though I am not sure that the parallel is perfect to Queller’s egalitarian/fraternal transitions, which I thought was more an issue of whether relatedness would or would not play a role in the evolution of cooperation.)

General comments:

The supplemental tables are extremely hard to dissect (mostly look to be pasting directly from statistical output) and the supplemental table legends often do not provide enough detail about what the values are to make any sense of them. The text frequently refers to the Supplemental Tables for support, so making the contents of these tables more clear is important.

The next-to-last paragraph is extremely difficult to follow. I think the authors are trying to get at the issue of whether flocculation would exclude non-relatives and if so how it would be accomplished (discriminating based on phenotype or genotype.) This is an important point because it goes to the heart of the distinction between “staying together” and “coming together”, as the latter can involve interactions among unrelated individuals but the former usually does not. However, I don’t think the authors know the answer for their system, and I suspect this point will be lost on most of the readership. At the very least, as written it is very hard to make sense of, and I would consider re-writing and making it more succinct.

Minor comments:

End of abstract: “and influence subsequent multicellular evolution”? I don’t see how subsequent multicellular evolution was influenced in these experiments. (Note also typo in next-to-last sentence; verb “observe” is repeated.)

Figure 1E. Only one test statistic presented for two different tests, inflection point and maximum growth rate. The provided t-test does not seem to match the data in the Supplemental Table.

Figure 1E. Snowflakes – why are there 11 points on the graph? Should be only 10?

Figure 2A. In my print version, it looks like the symbols have been pasted on top of the graph and they have shifted around. (It’s unclear whether it is a mistake or not, because as written I don’t know how many points to expect on the graph.) The dotted trendline, representing the results when snowflakes invade from rare, falls at or below all of the squares, which is odd for a trendline (they usually, by definition, are placed intermediate to the actual points.) Some attention needs to be given to this figure.

Figure 2B. Was a single snowflake and single unicell isolated from each of the 10 evolved populations and then used in these invasion assays? It is not clear what strains’ fitness data form the basis of the geometric means.

Figure 6B. In general, I like this figure, but I am confused by the floc- and floc+ (and some

of the related text). I thought the results ultimately suggested that both the ancestral and derived unicells were both flocculating (even though it only beneficially impacted settling in the presence of SFs), so why is there a mix of floc+ and floc- cells shown for social aggregation? Or is this hypothetical, demonstrating that any floc- strains, should they arise, would be excluded? Similarly, why are the snowflakes drawn smooth, but then shown as flocculating in panel C? If A-C are meant to illustrate a general concept, the figure should really be labeled as a schematic. (I see that panel D presents results.)

I find the comment that groups that form by staying together “purge genetic variability” an odd thing to say. It seems more like they don’t have any genetic heterogeneity because of how they form by cell division.

Reviewer #1 (Remarks to the Author):

As described in my previous review, this is an interesting paper that is now further strengthened by addressing my concerns in the prior review with new experiments. In particular, the additional experiments regarding “stickiness” vs. “cooperative flocculation” significantly improve the impact of the results. This was also a highlight weakness by reviewer #3.

However, prior to publication, the authors should consider the following set of revision. The manuscript in its current state is a bit challenging to read and understand the rationale. The introduction in particular needs some attention for clarity. The following suggestions for revision would significantly improve the quality and impact of the manuscript.

In particular, the authors could address a concern of reviewer #2 (“Other points The authors argue they are testing the hypothesis that their previous evolution of multicellularity in *S. cerevisiae* was due to whole genome duplication in this part of the hemiascomycetes by asking whether *K. lactis* can evolve multicellularity. There are two problems with this: 1) the hypothesis is about as idle as speculation can get and 2) whether *K. lactis* can or cannot easily evolve multicellularity has no bearing on the hypothesis, unless the authors are prepared to determine the genetic trajectories that lead to the phenotype in the two organisms.”) simply by revising the introduction and rationale for performing selection on *K. lactis*. The paragraph in the intro setting up the paper as it currently does makes it difficult to read and understand.

We have revised the Introduction accordingly (detailed explanations below).

Otherwise, this is a great paper with just a few rough edges that need to be worked out before publication.

Text revisions for clarity

Line 35-36, this part of the sentence should be deleted to keep the intro on point:

as exemplified by cancer², social parasitism³, and meiotic drive⁴

We have revised the text as suggested.

Line 36, delete “interdisciplinary”

We have revised the text as suggested.

Line 40, revise “According to this theory...” to “According to kin selection, the cost of cooperation can be offset by benefits to relatives, which are more likely to share alleles associated with cooperation”.

We have revised the text as suggested.

Line 44-45, revise to read “The origin and maintenance of cooperation among formerly free-living unicells is often viewed as a barrier for the evolution of complex multicellular life^{1,9,10}”

We have revised the text as suggested.

Line 47, revise to read “suggesting that solutions to evolving cooperation have evolved multiple independent times¹¹”

We have revised the text as suggested.

Line 52-54, revise to read “Nevertheless, cooperation among non-relatives has been documented multiple times^{16,17}, including altruism during the development of multicellular chimeras¹⁸.”

We have revised the text as suggested.

Line 54-57, this sentence is confusing are the authors referring to selection or that these sells are “selective” in who they cooperate with. It is at this point that the authors introduce a new term for the manuscript “selectivity”. I think selectivity makes this part hard to read. Revise to read “These observations suggest that other factors can limit the advantages of genetic similarity for cooperation, though measuring the cost and benefits of cooperativity is challenging.”

We have removed the confusing term, and revised this paragraph to improve readability. (In the course of that revision, we also removed the referenced sentence about ‘other factors’.)

The introduction should have a brief introduction to sociality regarding multicellularity. This is a critical oversight. There is discussion of kin selection, but a brief discussion of clonal vs. social multicellular mechanisms would be useful (3-4 sentences). E.g. summarize and refer to Grosberg 2007.

We have revised the text as suggested, including a brief introduction of the alternative paths to multicellularity (although we stay with the terms we use throughout, coming together and staying together). We also cite the suggested reference.

A brief introduction to *K. lactis* and its floc would be useful.

We have replaced the previous overview of differences between S. cerevisiae and K. lactis (e.g. the whole genome duplication) with a brief introduction of flocculation in budding yeast.

Lines 65-75 needs revision. There are a lot of ideas packed into this paragraph that are not fully completed.

We have rewritten this paragraph.

Lines 78-81 need revision. The set up for the series of experiments in these two sentences make the paragraph hard to read. My suggestions is “The goal of this work is to understand the outcome selection by sedimentation on K. lactis, an organism with conditional cooperative flocculation, compared to the outcome of S. cerevisiae which is not flocculant. Factors such as could also influence the outcome of the experiment”.

We have revised the text largely along the lines suggested, with relatively slight modifications.

As this paragraph currently reads, the paper appears to be much less interesting than it actually is. It is not until line ~180 that the idea of K. lactis being flocculant versus non-flocculant is introduced. I would frame the manuscript in terms of the interesting bit. The rest of the notable differences to S. cerevisiae and K. lactis noted here are a bit of red herring. The experimental test that is being performed here is K. lactis flocc vs. non-flocc. Reviewer #2 criticizes the paper for focusing on these as possible reasons for differences between strains. This paragraph really needs to be revised to address reviewer #2's concerns as well as make the manuscript more understandable as to what is really being tested here.

We have revised the text as suggested with respect to these concerns (although see Reviewer #2's response to our previous changes).

Line 202, there is a stray period.

We have revised the text as suggested.

Lines 215-217, change “, which” to “that”. The sentence itself should be revised for clarity.

We have revised the text as suggested.

Lines 217-222, revise to read “Snowflake development necessarily excludes non-kin from clusters via frequent genetic bottlenecks imposed by daughter cells staying together following division³⁴. Why does aggregation with non-kin (including possible cheaters) persist in lineages that already form multicellular clusters with close kin?”

We have revised the text as suggested.

Line 223, revise to read "This question was addressed by altering flocculation and determine the effect on settling behavior".

We have revised the text as suggested.

Reviewer #2 (Remarks to the Author):

This is clearly going to be an editorial decision. The glass half full view is that the authors have made a detailed response to every point made by the reviewers, agree with many of the points they raise, clarified their writing, and performed new experiments. The glass half empty view is that in many cases the detailed response is an argument for not doing an experiment, and in several cases, such as trying to disrupt flocs while they are alive, the experiment is one that has been suggested by multiple reviewers.

We will respond to the specific criticisms below, but note that the first revision of the manuscript included results demonstrating that chemical disruption of flocs reduces settling in both live (Figure 5) and heat-killed (Figure S2 (formerly Figure S1)) K. lactis cultures.

Responses to points I raised

"This paper builds on earlier work from Trivisano, which showed that selection for rapid settling produced multicellular aggregates that arose because mother and daughter cells failed to separate after cytokinesis. The new features of this work repeating the selection in Kluyveromyces lactis, showing that in this organisms the mixtures of multicellular aggregates and single cells exhibit frequency-dependent selection (whoever is in the minority has the selective advantage) and arguing that the K. lactis shows the maintenance of two phenotypes because sticky, single cells can settle faster by sticking to clumps and clumps can gain a similar advantage by sticking to each other. The authors clothe these experiments in the provocative title "Synergistic cooperation promotes multicellular performance and unicellular free-rider persistence", but Desai has already published a nice paper showing frequency-dependent selection between two budding yeast populations that have subtly different life styles, which included a mathematical analysis of the phenomenon, (PNAS 112, 11306), the idea and observation that stickiness reduces the selective difference between clumps and single cells seems unremarkable, and I am unconvinced about the arguments about group selection."

The authors argue that their work makes two fundamental contributions: 1) it shows cooperative interactions between non-relatives, 2) it allows the selection of cheats that do not destroy the original cooperation. My concern is that both of these phenomena are specialized consequences of the particular selection for faster sedimentation, and that neither result would be generalizable to more traditional forms of cooperation, such as the collective use of public goods. To me this is a reflection of the overemphasis on "levels of selection" and exception to rules that excites a certain section of evolutionary biologists.

We disagree with this assessment and address each part in turn:

“...both of these phenomena are specialized consequences of the particular selection for faster sedimentation...”

1. ***The same sedimentation selection regime produced divergent results in two different budding yeast.*** A central point in our manuscript is that unicellular free-riding and coexistence in the present study (*K. lactis*) are departures from previous evolution experiments that applied the same sedimentation selection regime (down to and including the growth medium) to *S. cerevisiae*. It is therefore hard to imagine how these results could be viewed as ‘specialized consequences’ of an approach that did not previously produce either of these same results (free-riding unicells and multi-snowflake cooperation) in a different species.

“...neither result would be generalizable to more traditional forms of cooperation, such as the collective use of public goods...”

2. ***Coexistence of public goods producers and non-producers is frequently observed in microbial populations.*** Our findings that cooperation occurs among non-relatives and survives cheating/free-riding are consistent with several other studies that have demonstrated coexistence between public goods producers and non-producers in microbial systems. Prominent examples include invertase in yeast (in structured¹ and unstructured² environments) and secreted iron-scavenging siderophores in *E. coli*³. A similar pattern has been demonstrated in natural populations of marine bacteria (which harbor siderophore producers and free-riding non-producers⁴), and although the same caliber of experiments have yet to be conducted, there is evidence that toxins produced by micro-algae can benefit nontoxic free-riders during blooms^{5,6}. Our finding that free-riders stably coexist with cooperators is novel in that it does represent a substantial departure from previous evolution experiments with incipient multicellular populations (which have either found that cheaters destroy cooperation^{7,8}, or that interactions with non-relatives are simply precluded by multicellular development⁹⁻¹¹), however, there are ready parallels if one considers microbial cooperation more broadly.

*The authors argue they are testing the hypothesis that their previous evolution of multicellularity in *S. cerevisiae* was due to whole genome duplication in this part of the hemiascomycetes by asking whether *K. lactis* can evolve multicellularity. There are two*

problems with this: 1) the hypothesis is about as idle as speculation can get and 2) whether K. lactis can or cannot easily evolve multicellularity has no bearing on the hypothesis, unless the authors are prepared to determine the genetic trajectories that lead to the phenotype in the two organisms.

The authors' response is satisfactory.

Phrases like this "Unlike 'directed evolution,' this protocol does not select for specific traits other than fitness; rather, it imposes selection and evolutionary responses are observed. " are profoundly unhelpful: 1) without defining "directed evolution" and citing references, the reader has no idea who is being attacked, 2) many, many studies in experimental evolution select for fitness as faster proliferation, which is no more directed than the selection the authors use, and 3) selecting for how fast cells fall to the bottom of a tube would seem to many to be a somewhat contrived and artificial selection, and the argument that this is a proxy for various forms of more natural natural selection (the duplication is intentional) is unconvincing.

The authors' address the first two points and argue that "How artificial is 'too' artificial is ultimately determined on the questions being asked, and we believe settling selection is justifiable from the perspective of the questions we address here." The difference between their position and mine is a reflection of fundamental differences about how, in evolutionary questions, one should argue from the concrete to the abstract and cannot be easily resolved.

We accept that we won't be able to convince the reviewer on this particular point.

There is a simple experiment to test the idea that stickiness is what leads to the persistence of unicellular cells: varying the density at which cultures are propagated. If the single cells persist in cultures subject to settling selection, their ability to do so will depend on culture density, since collisions between clumps and single cells are required to settle single cells, and the rate of collision will go as the product of the density of the two genotypes. This would give rise to exactly the same sort of mathematical analysis that Desai used to demonstrate that predictions of their model for coexistence were quantitatively met. Another useful experiment would be to control the degree of stickiness in S. cerevisiae and show that this could recapitulate the findings in K. lactis.

The authors ignore the second suggestion and argue against the first, claiming that dilution will affect more than the collision between single cells and snowflakes. My position is that as long as the experiment is performed under conditions where snowflakes settle, whether they stick to each other or not, the effect of dilution should be to reduce the benefit to single cells, and that showing this to be true would go a long way to support the model.

We have now conducted new experiments to address this concern. Our efforts have focused on the 'dilution' approach, rather than the suggestion to reproduce K. lactis-like flocculation in S. cerevisiae. (The reviewer notes that we did not address this latter suggestion in our first

response. There are several likely complications with the idea of reproducing K. lactis-like flocculation in S. cerevisiae; for instance, floc proteins of Kluyveromyces and Saccharomyces are highly diverged and recognize different carbohydrates (galactose and mannose, respectively) at the ‘target’ cell surface, so there is no guarantee that a K. lactis floc protein expressed in S. cerevisiae would induce similar degrees of flocculation among S. cerevisiae cells. Furthermore, relatively little is known about floc regulation in K. lactis, and preliminary gene expression data suggests dynamic regulation throughout the growth cycle (which differs across ancestral and evolved K. lactis). Teasing apart these mechanistic issues in K. lactis itself is already proving far from trivial.)

Our interpretation of the suggested experiment is that it is intended to test the hypothesis that unicell settling depends on collisions between unicells and snowflakes. Towards this end, we first tested a range of unicell and snowflake densities (i.e. we independently diluted each phenotype) in a factorial experiment, then performed a triplicated experiment focused on the most informative factor combinations.

The results of these experiments unambiguously support the expectation that unicell performance during settling experiments increases with snowflake density in relatively sparse cultures, i.e. those in which unicell performance is expected to be limited by settling. However, as we mentioned previously, in dense snowflake cultures, the volume of settled material greatly exceeds (by about 2-fold) the volume that is selectively passaged to the next day’s growth medium. As a result, diluting snowflakes has a quadratic effect on unicell performance across all densities, including strong and highly significant positive linear and negative quadratic terms ($p < 0.0001$; Table S4, Figure S1). We further found no significant effect of increasing unicell density ($p = 0.81$), consistent with the view that unicells are ‘free-riders’ that do not contribute to improved settling.

To summarize: from low to moderate snowflake densities, these new results clearly support the prediction identified by the reviewer:

“(T)he effect of dilution should be to reduce the benefit to single cells, and ... showing this to be true would go a long way to support the model”.

Reviewer #3 (Remarks to the Author):

Overall I am satisfied with the revisions, and most of my questions have been answered. I like the “coming together” or “staying together” vocabulary, and I think it is a very useful distinction (though I am not sure that the parallel is perfect to Queller’s egalitarian/fraternal transitions, which I thought was more an issue of whether relatedness would or would not play a role in the evolution of cooperation.)

General comments:

The supplemental tables are extremely hard to dissect (mostly look to be pasting directly from statistical output) and the supplemental table legends often do not provide enough detail about what the values are to make any sense of them. The text frequently refers to the Supplemental Tables for support, so making the contents of these tables more clear is important.

We have extensively revised the supplemental tables for accessibility and presentation.

The next-to-last paragraph is extremely difficult to follow. I think the authors are trying to get at the issue of whether flocculation would exclude non-relatives and if so how it would be accomplished (discriminating based on phenotype or genotype.) This is an important point because it goes to the heart of the distinction between “staying together” and “coming together”, as the latter can involve interactions among unrelated individuals but the former usually does not. However, I don't think the authors know the answer for their system, and I suspect this point will be lost on most of the readership. At the very least, as written it is very hard to make sense of, and I would consider re-writing and making it more succinct.

We have rewritten the final two paragraphs of the Discussion section to be more accessible and to state more clearly how the present study bears on broader questions pertaining to the evolution of multicellularity.

Minor comments:

End of abstract: “and influence subsequent multicellular evolution”? I don't see how subsequent multicellular evolution was influenced in these experiments. (Note also typo in next-to-last sentence; verb "observe" is repeated.)

We have revised the text as suggested.

Figure 1E. Only one test statistic presented for two different tests, inflection point and maximum growth rate. The provided t-test does not seem to match the data in the Supplemental Table.

We have revised the text as suggested.

Figure 1E. Snowflakes – why are there 11 points on the graph? Should be only 10?

The reviewer is correct: a superfluous snowflake isolate (which was not used at any other point in the experiments reported in this manuscript) was mistakenly included in this plot. We have removed the extra point and updated the confidence ellipse accordingly, and thank the reviewer for the keen eye.

Figure 2A. In my print version, it looks like the symbols have been pasted on top of the graph

and they have shifted around. (It's unclear whether it is a mistake or not, because as written I don't know how many points to expect on the graph.) The dotted trendline, representing the results when snowflakes invade from rare, falls at or below all of the squares, which is odd for a trendline (they usually, by definition, are placed intermediate to the actual points.) Some attention needs to be given to this figure.

We have extensively revised this figure for improved clarity. We concede that the earlier form was counter-intuitive, because the symbols showed frequencies only after settling selection, whereas the lines were based on average frequencies (based on separate measures before and after selection). In the revised figure, the (far fewer) points represent the average of frequencies before and after selection and are therefore consistent with the line. (We have also expanded Figure 2A from showing a single 'example' population (previously) to showing averages across all ten replicate populations for each initial state.)

Figure 2B. Was a single snowflake and single unicell isolated from each of the 10 evolved populations and then used in these invasion assays? It is not clear what strains' fitness data form the basis of the geometric means.

Yes, each of the 10 replicate populations is represented by a single unicell and snowflake isolate in these experiments. We have revised the figure legend to clarify this point.

Figure 6B. In general, I like this figure, but I am confused by the floc- and floc+ (and some of the related text). I thought the results ultimately suggested that both the ancestral and derived unicells were both flocculating (even though it only beneficially impacted settling in the presence of SFs), so why is there a mix of floc+ and floc- cells shown for social aggregation? Or is this hypothetical, demonstrating that any floc- strains, should they arise, would be excluded? Similarly, why are the snowflakes drawn smooth, but then shown as flocculating in panel C? If A-C are meant to illustrate a general concept, the figure should really be labeled as a schematic. (I see that panel D presents results.)

We have revised this figure to avoid the confusion stemming from the inconsistent use of cell outlines. We now use different colors (blue and red) to denote snowflake and floc, respectively (e.g. a floc+ unicell is red, a floc+ snowflake is purple).

I find the comment that groups that form by staying together "purge genetic variability" an odd thing to say. It seems more like they don't have any genetic heterogeneity because of how they form by cell division.

We have revised the text as suggested.

REVIEWERS' COMMENTS:

Reviewer #1 (Remarks to the Author):

The authors did an exceptional job in addressing all criticisms. The current version of the manuscript is a considerable improvement on the initial submission.

I am satisfied with the author's response to my prior corrections and queries, as well as those criticisms raised by Reviewer #2 in particular.